# $K$-Nearest-Neighbor Local Sampling Based Conditional Independence Testing

**Shuai Li, Yingjie Zhang**
School of Statistics
KLATASDS-MOE
East China Normal University

**Hongtu Zhu**
Departments of Biostatistics,
Statistics, Computer Science, and Genetics
The University of North Carolina at Chapel Hill

**Christina Dan Wang**
Business Division
New York University Shanghai

**Hai Shu**
Department of Biostatistics
School of Global Public Health
New York University

**Ziqi Chen**,[*] **Zhuoran Sun, Yanfeng Yang**
School of Statistics
KLATASDS-MOE
East China Normal University

## Abstract

Conditional independence (CI) testing is a fundamental task in statistics and machine learning, but its effectiveness is hindered by the challenges posed by high-dimensional conditioning variables and limited data samples. This article introduces a novel testing approach to address these challenges and enhance control of the type I error while achieving high power under alternative hypotheses. The proposed approach incorporates a computationally efficient classifier-based conditional mutual information (CMI) estimator, capable of capturing intricate dependence structures among variables. To approximate a distribution encoding the null hypothesis, a $k$-nearest-neighbor local sampling strategy is employed. An important advantage of this approach is its ability to operate without assumptions about distribution forms or feature dependencies. Furthermore, it eliminates the need to derive asymptotic null distributions for the estimated CMI and avoids dataset splitting, making it particularly suitable for small datasets. The method presented in this article demonstrates asymptotic control of the type I error and consistency against all alternative hypotheses. Extensive analyses using both synthetic and real data highlight the computational efficiency of the proposed test. Moreover, it outperforms existing state-of-the-art methods in terms of type I and II errors, even in scenarios with high-dimensional conditioning sets. Additionally, the proposed approach exhibits robustness in the presence of heavy-tailed data.

## 1 Introduction

Testing for conditional independence (CI) is a crucial and challenging task in statistics and machine learning, with wide-ranging applications in graphical models [26, 17], causal inference [44, 37, 22], and variable selection [11, 24]. The objective is to determine whether two random variables, $X$ and $Y$, are independent given a set of conditioning variables $Z$, based on observations of the joint

---

[*]Corresponding author (zqchen@fem.ecnu.edu.cn).

37th Conference on Neural Information Processing Systems (NeurIPS 2023).

distribution $p_{X,Y,Z}(x, y, z)$. Specifically, the hypothesis to be tested is:

$$H_0 : X \perp\!\!\!\perp Y | Z \quad \text{versus} \quad H_1 : X \not\!\perp\!\!\!\perp Y | Z. \tag{1}$$

However, CI testing becomes challenging due to the high-dimensionality of the conditioning variables $Z$, and the limited availability of data samples [5, 35, 40, 1]. Existing tests may struggle to control the type I error, particularly when handling high-dimensional conditioning variable sets with complex dependency structures [6, 40]. Moreover, even when a test is valid, the limited data availability can make it exceedingly challenging to distinguish between null and alternative hypotheses, resulting in low testing power [40].

In this article, we present a novel conditional independence (CI) testing method based on a $k$-nearest-neighbor local sampling scheme. Our approach incorporates two essential components: a conditional mutual information (CMI) estimator utilizing classification techniques, and a $k$-nearest-neighbor local sampling strategy to approximate the conditional distribution $p_{X|Z}(x|z)$. This approximation enables us to simulate data sets from a distribution that represents the null hypothesis, allowing us to estimate the CMI using these simulated data sets. By comparing the CMI estimator based on real data with those computed from the simulated data sets, we make informed decisions regarding the hypothesis testing. Theoretical analysis demonstrates that our proposed test achieves a valid control of the type I error asymptotically and exhibits consistency against all alternatives in $H_1$. Synthetic and real data analyses showcase that our test outperforms previous methods in terms of both the type I error and power under $H_1$, even when dealing with high-dimensional conditioning sets and/or small datasets. Furthermore, our approach remains computationally efficient as the dimension of the conditioning set and/or sample size increase. Additionally, our method is robust even in the presence of heavy-tailed data.

## 2 Related Work

### 2.1 CMI estimation

In our work, we utilize CMI as a measure of conditional dependence. CMI offers a strong theoretical guarantee for CI testing, where $I(X; Y|Z) = 0 \iff X \perp\!\!\!\perp Y|Z$ [14]. It has the ability to capture complex dependence structures among variables, even in non-linear scenarios. Estimators based on $k$-nearest neighbors and kernel methods have been widely employed for CMI estimation [27, 38, 23, 19]. However, these methods may encounter efficiency issues when dealing with high-dimensional conditioning variables, known as the curse of dimensionality [34, 32]. [32] approached the CMI estimation problem by formulating it as a minimax optimization problem and proposed the use of generative adversarial networks (GANs) to optimize and obtain the CMI estimator, which can handle high-dimensional scenarios. However, the training of GANs is often challenging, with the risk of collapse if hyperparameters and regularizers are not carefully chosen [15]. Recently, [34, 30] proposed a classifier-based CMI estimator capable of handling high dimensions. They initially developed an estimator for the Kullback-Leibler (KL) divergence using a classifier and then derived mutual information (MI) estimators based on it. CMI is defined as the difference between two MI values, and the CMI estimator is obtained by computing the difference between two MI estimators. In contrast, our method directly utilizes the classifier-based KL-divergence estimator to obtain the CMI estimator. This approach also handles high dimensions effectively and is computationally more efficient compared to the method employed by [34, 30].

### 2.2 CI Testing

In recent years, a considerable body of literature on CI testing has emerged. Here, we provide a brief overview of some existing methods, and for a more comprehensive review, we refer readers to [28].

One important category of CI testing methods involves using different measures for CI [3, 46, 2]. For example, [18] proposed a CI testing method based on the empirical Hilbert-Schmidt norm of the conditional cross-covariance operator. Another approach, KCIT, was introduced by [47], which utilizes the partial association of reproducing kernel Hilbert spaces to measure conditional independence. [40] proposed a novel kernel-based CI testing method using the $l_p$ distance between two well-chosen analytic kernel mean embeddings evaluated at a finite set of locations. [42] proposed the generalized covariance measure for CI testing based on a regression method. However, obtaining the exact distribution of the test statistic derived from the conditional independence measure under

$H_0$ can be challenging. Instead, researchers have employed alternative methods to derive asymptotic distributions of the test statistics under the null hypothesis [47, 46, 45, 40]. Nevertheless, the effectiveness of asymptotic distributions can be compromised when the sample size is small or the dimension of $Z$ is high [16, 38]. As a result, tests based on asymptotic distributions may exhibit inflated type-I errors or inadequate power when dealing with small sample sizes or high-dimensional $Z$ [47, 40]. In our work, we choose to utilize CMI as a metric for conditional dependence.

To overcome the reliance on asymptotic null distributions, [11] proposed the model-X framework, assuming knowledge of the conditional distribution $p_{X|Z}(x|z)$. Under this assumption, a set of test statistics can be computed that are exchangeable under the null hypothesis, either through direct resampling [11] or permutation methods [8]. However, in practice, the distribution of $X|Z$ is rarely available. Therefore, accurately approximating the distribution of $X|Z$ becomes crucial for maintaining the type I error control of these tests. To address this challenge, [5] developed the generative conditional independence test (GCIT) using Wasserstein GANs to approximate the distribution of $X|Z$, while [43] proposed using Sinkhorn GANs for the same purpose. However, as highlighted in [5, 30], limited data and noise may lead to inaccurate learning of conditional distributions using neural networks, resulting in inflated type I errors for GCIT and the double GANs-based CI test (DGCIT). To mitigate this issue, [30] proposed using the 1-nearest neighbor method to generate samples from the approximated conditional distribution of $X$ given $Z$. However, their method requires splitting the dataset into two parts, with the testing dataset used for calculating the test statistics comprising only one-third of the total samples. This will reduce the statistical power of the test, particularly when working with small datasets. [38] utilized a Kozachenko-Leonenko estimator for CMI as the test statistic and proposed a permutation scheme based on $k$-nearest neighbors to generate samples under the null distribution. However, the curse of dimensionality adversely affects the performance of the Kozachenko-Leonenko CMI estimator, leading to poor performance when the conditioning variable set $Z$ has high dimensions. Additionally, no theoretical guarantee is provided. [31] and [25] employed a binning strategy, discretizing $Z$ into a finite number of bins based on the proximity of conditioning variables $Z$, followed by the "permute-within-groups" strategy. However, selecting bins in high-dimensional settings presents a significant challenge [8], and their methods are limited to handling conditioning variables $Z$ with very few dimensions.

In our work, we propose the utilization of a $k$-nearest-neighbor local sampling strategy as an appealing alternative to the binning strategy. Specifically, our strategy generates samples locally based on the $k$-nearest neighbors of the conditioning variables $Z$. We demonstrate that the distribution of samples generated from this $k$-nearest-neighbor local sampling scheme closely approximates the true conditional distribution $p_{X|Z}(x|z)$ in terms of total variation distance. One significant advantage of our proposed method is that it allows the entire dataset to be used for computing the testing statistics, eliminating the need for dataset splitting. This feature makes our method more effective compared to [30], which requires dataset splitting, particularly when dealing with small datasets. Moreover, our method does not require the derivation of asymptotic null distributions for the estimated CMI, and it can easily handle high-dimensional conditioning variables $Z$. We provide theoretical and empirical evidence that our test achieves a valid control of the type I error and attains high power under the alternative hypothesis $H_1$.

## 3 $K$-Nearest-Neighbor Local Sampling Based CI Testing

### 3.1 Classifier-based CMI estimator

The CMI for a triplet of random variables/vectors $(X, Y, Z)$ is defined as:

$$I(X; Y|Z) = \iiint p_{X,Y,Z}(x, y, z) \log \frac{p_{X,Y,Z}(x, y, z)}{p_{X,Z}(x, z)p_{Y|Z}(y|z)} dx dy dz, \qquad (2)$$

where $p_{X,Y,Z}(x, y, z)$ is the joint density of $(X, Y, Z)$, $p_{X,Z}(x, z)$ is the joint density of $(X, Z)$, and $p_{Y|Z}(y|z)$ is the conditional density of $Y$ given $Z = z$. One approach to estimate CMI is by directly estimating the joint and conditional densities from the available data and plugging them into (2). However, accurately estimating the density functions can be challenging, especially in high-dimensional settings where it is more difficult than directly estimating CMI (2) [34]. CMI is a special case of the Kullback-Leibler (KL) divergence, and thus we have:

$$I(X; Y|Z) = D_{KL}(p_{X,Y,Z}(x, y, z) || p_{X,Z}(x, z)p_{Y|Z}(y|z)), \qquad (3)$$

**Algorithm 1** 1-Nearest-Neighbor sampling (**1-NN**$(V_1, V_2, n)$)

---

**Input**: Datasets $V_1$ and $V_2$, both with sample size $n$ and $V = V_1 \cup V_2$ consisting of $2n$ independently and identically distributed (i.i.d.) samples from $p_{X,Y,Z}(x, y, z)$.
**Output**: A new data set $V'$ consists of $n$ samples.

1: Let $V' = \emptyset$.
2: **for** $(X, Y, Z)$ in $V_2$ **do**
3:   Go to $V_1$ to find the sample $(X', Y', Z')$ such that $Z'$ is the 1-nearest neighbor of $Z$ in terms of the $l_2$ norm
4:   $V' = V' \cup \{(X, Y', Z)\}$.
5: **end for**
6: **return** $V'$

---

where $D_{KL}(f||g)$ denotes the KL divergence between two distribution functions $F$ and $G$, with density functions $f(x)$ and $g(x)$, respectively. The Donsker-Varadhan (DV) representation of $D_{KL}(f||g)$ is given by:

$$\sup_{s \in \mathcal{S}} \left[ E_{w \sim f} s(w) - \log E_{w \sim g} \exp\{s(w)\} \right], \tag{4}$$

where the function class $\mathcal{S}$ includes all functions with finite expectations. The optimal function in (4) is given by $s^*(x) = \log\{f(x)/g(x)\}$ [4], which leads to:

$$D_{KL}(f||g) = E_{w \sim f} \log\{f(w)/g(w)\} - \log[E_{w \sim g}\{f(w)/g(w)\}]. \tag{5}$$

Building upon the classifier-based CMI estimation method proposed by [34], we propose a CMI estimation method using the 1-nearest-neighbor (1-NN) sampling algorithm [41, 30]. The pseudocode for estimating CMI is outlined in Algorithm 2. The main objective is to empirically estimate (5) with $f = p_{X,Y,Z}(x, y, z)$ and $g = p_{X,Z}(x, z)p_{Y|Z}(y|z)$, which requires samples from both $p_{X,Y,Z}(x, y, z)$ and $p_{X,Z}(x, z)p_{Y|Z}(y|z)$. The available data only consists of samples from $p_{X,Y,Z}(x, y, z)$. However, generating samples from $p_{X,Z}(x, z)p_{Y|Z}(y|z)$ requires knowledge of the unknown conditional distribution $p_{Y|Z}(y|z)$. To address this challenge, we propose utilizing the 1-NN sampling algorithm [41, 30], which is outlined in Algorithm 1. Empirical and theoretical results presented in [30] indicate that the 1-NN sampling algorithm can accurately approximate the conditional distribution.

Next, we formalize the classifier-based CMI estimator. We consider a data set $V$ consisting of $2n$ i.i.d. samples $\{W_i := (X_i, Y_i, Z_i)\}_{i=1}^{2n}$ with $(X_i, Y_i, Z_i) \sim p_{X,Y,Z}(x, y, z)$. The data set $V$ is divided into two equally sized parts $V_1$ and $V_2$, where $|V_1| = |V_2| = n$. For data sets $V_1$ and $V_2$, we use the 1-NN sampling algorithm 1 to generate a new data set $V'$ with $n$ samples. We assign labels $l = 1$ for all samples in $V_2$ and $l = 0$ for all samples in $V'$. In this supervised classification task, a binary classifier can be trained using an advanced binary classification model, such as XGBoost [41, 12] or deep neural networks [21]. The classifier produces predicted probability $\alpha_m = P(l = 1|W_m)$ for a given sample $W_m$, leading to an estimator of the likelihood ratio on $W_m$ given by $\widehat{L}(W_m) = \alpha_m/(1 - \alpha_m)$. It follows from (3) and (5) that an estimator of $I(X; Y|Z)$ is given by

$$\widehat{I}(X; Y|Z) := \widehat{D}_{KL}(p_{X,Y,Z}(x, y, z)||p_{X,Z}(x, z)p_{Y|Z}(y|z))$$

$$= d^{-1} \sum_{i=1}^{d} \log \widehat{L}(W_i^f) - \log\{d^{-1} \sum_{j=1}^{d} \widehat{L}(W_j^g)\}, \tag{6}$$

where $d = \lfloor n/3 \rfloor$ with $\lfloor t \rfloor$ being the largest integer not greater than $t$, $W_i^f$ is a sample in $V_f^{\text{test}}$ and $W_j^g$ is a sample in $V_g^{\text{test}}$, where $V_f^{\text{test}}$ and $V_g^{\text{test}}$ are defined in Algorithm 2. According to Theorem 1 in [34], $\widehat{I}(X; Y|Z)$ is a consistent estimator of $I(X; Y|Z)$.

In contrast to the approach taken by [34, 30], which estimate CMI by using the difference between two mutual informations (i.e., $I(X; Y|Z) = I(X; Y, Z) - I(X; Z)$) and therefore require training two binary classifiers, our method achieves CMI estimation with just a single binary classifier. This leads to significantly improved computational efficiency. As depicted in Algorithm 3, calculating a single $p$-value requires the computation of $(B + 1)$ CMIs. Hence, in practical applications involving real data analysis, especially when dealing with large sample sizes or high-dimensional conditional variables, efficient computation of the CMI becomes crucial.

**Algorithm 2** Classifier-based CMI Estimator
___
**Input**: Dataset $V$ containing $2n$ i.i.d. samples drawn from $p_{X,Y,Z}(x, y, z)$.
**Output**: An estimator of CMI.
1: Equally split $V$ into two parts $V_1$ and $V_2$, each containing $n$ samples.
2: Apply Algorithm 1 to generate a new dataset $V'$ with $|V'| = n$.
3: Form the labeled datasets $V_f = \{(W_i^f, l = 1) : W_i^f \in V_2\}$ and $V_g = \{(W_j^g, l = 0) : W_j^g \in V'\}$.
4: Divide $V_f$ into training and testing subsets $V_f^{\text{train}}$ and $V_f^{\text{test}}$, at a ratio of 2:1.
5: Similarly, split $V_g$ into training and testing subsets $V_g^{\text{train}}$ and $V_g^{\text{test}}$, at a ratio of 2:1.
6: Merge the datasets to form $V^{\text{train}} = V_f^{\text{train}} \cup V_g^{\text{train}}$ and $V^{\text{test}} = V_f^{\text{test}} \cup V_g^{\text{test}}$.
7: Train the classifier $C$ using $V^{\text{train}}$.
8: For each $w \in V_0^{test}$, where $V_0^{test}$ includes all features in $V^{\text{test}}$, compute the classifier-based predicted probability $P(l = 1|w)$.
9: Calculate $\widehat{I}(X; Y|Z)$ as per formula (6).
10: **return** $\widehat{I}(X; Y|Z)$.
___

## 3.2 The Proposed CI Testing Procedure

While we have proposed a consistent estimator for CMI, accurately estimating it as zero under the null hypothesis, $H_0$, is practically unattainable due to random errors in the sample data. These errors lead to deviations of the estimator from the actual value. To enhance the effectiveness of CI testing based on the CMI estimator, we propose a test that takes into account the statistical variation inherent in the CMI estimator.

Under $H_0$, we can express the following representation:

$$p_{X|Y,Z}(x|y, z) = p_{X|Z}(x|z). \tag{7}$$

The null model in (7) preserves the dependence between $X$ and $Z$, while breaking any dependence between $X$ and $Y$. Therefore, if a direct causal link exists between $X$ and $Y$, replacing $X$ with a null sample $X^* \sim p_{X|Z}(x|z)$ would disrupt this relationship. Thus, we can conclude that $X^*$ and $Y$ are conditionally independent given $Z$, i.e., $I(X^*; Y|Z) = 0$.

Consider $n$ i.i.d. copies $\mathcal{A} = \{(x_i, y_i, z_i) : i = 1, \ldots, n\}$ of $(X, Y, Z)$. When the distribution of $X|Z$ is known, conditional on $\mathbf{Z} = (z_1, \ldots, z_n)^T$, we can independently draw a pseudo sample $x_i^{(b)} \sim p_{X|Z}(x|z_i)$ for each $i$ across $b = 1, \ldots, B$ such that all $\mathbf{X}^{(b)} := (x_1^{(b)}, \ldots, x_n^{(b)})^T$ are independent of $\mathbf{Y} := (y_1, \ldots, y_n)^T$ and also $\mathbf{X} := (x_1, \ldots, x_n)^T$, where $B$ is the number of repetitions. Under $H_0$, $(\mathbf{X}^{(b)}, \mathbf{Y}, \mathbf{Z}) \stackrel{d}{=} (\mathbf{X}, \mathbf{Y}, \mathbf{Z})$ for all $b$, where $\stackrel{d}{=}$ denotes equality in distribution. We denote the CMI estimator of $I(X; Y|Z)$ based on $(\mathbf{X}^{(b)}, \mathbf{Y}, \mathbf{Z})$ as $\widetilde{\text{CMI}}^{(b)}$ and denote the estimator based on $(\mathbf{X}, \mathbf{Y}, \mathbf{Z})$ as $\widehat{\text{CMI}}$. We can approximate the $p$-value by

$$p := \frac{1 + \sum_{b=1}^{B} \mathbf{1}(\widetilde{\text{CMI}}^{(b)} \geq \widehat{\text{CMI}})}{1 + B}, \tag{8}$$

where $\mathbf{1}(\cdot)$ is the indicator function. To begin with, we demonstrate that the test based on (8) can effectively control the type I error. Specifically, under the null hypothesis $H_0$, the $(B + 1)$ triples $(\mathbf{X}, \mathbf{Y}, \mathbf{Z}), (\mathbf{X}^{(1)}, \mathbf{Y}, \mathbf{Z}), \ldots, (\mathbf{X}^{(B)}, \mathbf{Y}, \mathbf{Z})$ are exchangeable, and thus the $p$-value is valid and satisfies $P(p \leq \alpha|H_0) \leq \alpha$ for any given $\alpha \in (0, 1)$ [11, 8, 43]. Furthermore, it intuitively suggests that our test can achieve high power under the alternative hypothesis $H_1$. As $n$ approaches infinity, $\widetilde{\text{CMI}}^{(b)}$ converges to zero in probability. But under $H_1$, we know that $I(X; Y|Z) > 0$. As a result, $\widehat{\text{CMI}}$ should be positive with high probability. Consequently, the $p$-value calculated using (8) is very small with high probability, indicating the consistency of our test against all alternatives stated in $H_1$.

However, $p_{X|Z}(x|z)$ is rarely known in practice [5, 43]. We propose using the $k$-nearest-neighbor local sampling strategy to approximate $p_{X|Z}(x|z)$. The approximated distribution is denoted as $\widehat{p}_{X|Z}(x|z)$. To generate pseudo samples from $\widehat{p}_{X|Z}(x|z)$, we follow the steps below:

**Algorithm 3** $K$-Nearest-neighbor local sampling based CI testing
___
**Input**: Dataset $(\boldsymbol{X}, \boldsymbol{Y}, \boldsymbol{Z})$ consisting of $n$ i.i.d. samples from $p_{X,Y,Z}(x, y, z)$.
**Parameter**: The number of repetitions $B$; the neighbor order $k$; the significance level $\alpha$.
**Output**: Accept $H_0 : X \perp\!\!\!\perp Y | Z$ or $H_1 : X \not\!\perp\!\!\!\perp Y | Z$.
___
1: Use Algorithm 2 to obtain $\widehat{\mathrm{CMI}}$ based on $(\boldsymbol{X}, \boldsymbol{Y}, \boldsymbol{Z})$.
2: **for** $i \in \{1, 2, ..., n\}$ **do**
3:  Obtain the set of indices of the $k$-nearest neighbor of $z_i$: $\mathcal{M}_i = \{j \in \{1, 2, ..., n\} : \|z_j - z_i\|_2 \leq d_i^k\}$, where $d_i^k$ denotes the distance of $z_i$ to its $k$-nearest neighbor.
4: **end for**
5: $b = 1$.
6: **while** $b \leq B$ **do**
7:  Initialize empty array $\widetilde{\boldsymbol{X}}$ of length $n$.
8:  **for** $i \in \{1, 2, ..., n\}$ **do**
9:    Shuffle $\mathcal{M}_i$
10:    $j = \mathcal{M}_i[1]$
11:    $\widetilde{\boldsymbol{X}}[i] = \boldsymbol{X}[j]$.
12:  **end for**
13:  Use Algorithm 2 to obtain $\widetilde{\mathrm{CMI}}^{(b)}$ based on $(\widetilde{\boldsymbol{X}}, \boldsymbol{Y}, \boldsymbol{Z})$.
14:  $b = b + 1$.
15: **end while**
16: Compute $p$-value: $p := \left[1 + \sum_{b=1}^{B} \mathbb{1}\{\widetilde{\mathrm{CMI}}^{(b)} \geq \widehat{\mathrm{CMI}}\}\right] / (1 + B)$.
17: **if** $p \geq \alpha$ **then**
18:  Accept $H_0 : X \perp\!\!\!\perp Y | Z$.
19: **else**
20:  Accept $H_1 : X \not\!\perp\!\!\!\perp Y | Z$.
21: **end if**
___

1. Obtain the set of indices of the $k$-nearest neighbors of $z_i$ based on the $l_2$ norm. Denote this set as $\mathcal{M}_i = \{j \in \{1, ..., n\} : \|z_j - z_i\|_2 \leq d_i^k\}$, where $d_i^k$ is the distance of $z_i$ to its $k$-nearest neighbor;

2. Shuffle $\mathcal{M}_i$ and let $j$ be the first element of $\mathcal{M}_i$. Then, define the pseudo sample $\tilde{x}_i$ as $x_j$;

3. Repeat the above two steps for all $i \in \{1, 2, \ldots, n\}$ to obtain the resultant pseudo sample vector $\widetilde{\boldsymbol{X}} = (\tilde{x}_1, \ldots, \tilde{x}_n)^T$.

We establish in Section 4 that the total variation distance between the true distribution of $X|Z$ and the distribution of samples generated by the $k$-nearest-neighbor local sampling strategy tends to zero in probability as $n$ goes to infinity. This evidence suggests that the latter is a close approximation of the former. Thus, $(\widetilde{\boldsymbol{X}}, \boldsymbol{Y}, \boldsymbol{Z})$ approximates directly a distribution that encodes the null hypothesis. We then compute the CMI estimator based on it. This process is repeated $B$ times, resulting in $B$ realizations of the CMI estimator under the null hypothesis denoted by $(\widetilde{\mathrm{CMI}}^{(1)}, \ldots, \widetilde{\mathrm{CMI}}^{(B)})$. We can determine whether to reject the null hypothesis by comparing these estimators to the one of the original sample set $\mathcal{A}$. Specifically, we calculate the $p$-value using (8), but with $\widetilde{\mathrm{CMI}}^{(b)}$ replaced by $\widetilde{\mathrm{CMI}}^{(b)}$. If the resulting $p$-value is smaller than the prespecified significance level, we reject the null hypothesis. The pseudo code is presented in Algorithm 3. In Section 4, we will demonstrate through theoretical analysis that our test asymptotically achieves a valid control of type I error and is consistent against all the alternatives in $H_1$.

[30] employed the 1-NN sampling strategy to approximate the distribution of $X|Z$. To ensure substantial dissimilarity between the pseudo-sample datasets generated across repetitions, their method requires dividing the dataset into two parts. As a consequence, the dataset used for calculating the test statistics consists of only one-third of the total samples. This limitation can result in reduced statistical power of the test, especially when working with small datasets. In contrast, our proposed procedure eliminates the need for dataset splitting and allows the entire dataset to be used in computing $\widehat{\mathrm{CMI}}$, thereby avoiding the loss of testing power.

# 4 Theoretical Results

In this section, we present our main theoretical results. All the detailed proofs for these results can be found in the Supplementary Materials. Let $P_1$ and $P_2$ be two probability distributions defined on the same probability space. The total variation distance between $P_1$ and $P_2$ is defined as $d_{TV}(P_1, P_2) = \sup_{A \subset \Omega} |P_1(A) - P_2(A)|$, where the supremum is taken over all measurable subsets $A$ of the sample space $\Omega$. We will show in Theorem 2 that the distribution of $\widetilde{X}$ generated by the $k$-nearest-neighbor local sampling strategy is very close to the true conditional distribution in terms of the total variation distance.

Lemma 1 plays a crucial role in proving Theorem 2. It demonstrates that the $k$-nearest neighbor of $Z$ has a similar property to the work of [13], where it has been shown that the nearest neighbor of $Z$ converges almost surely (a.s.) to $Z$ as the sample size $n$ approaches infinity.

**Lemma 1.** *Let $Z$ and $Z_1, \ldots, Z_n$ be i.i.d. random vectors from $p(z)$. For a given positive integer $k$, let $Z_n^{(k)}$ be the $k$-nearest neighbor of $Z$ from the set $\{Z_1, \ldots, Z_n\}$. Then, $Z_n^{(k)} \xrightarrow{a.s.} Z$ as $n \to \infty$.*

We make the following regularity conditions, which have been introduced in [19], [20] and [41]. To simplify the notation, we may drop the subscripts when referring to probability density functions.

**Assumption 1 (Smoothness on $p(x|z)$).** For any $z \in \mathbb{R}^{d_z}$ and any $a$ such that $\|a - z\|_2 \le \epsilon$, we assume $0 \le \lambda_{\max}(I_a(z)) \le \beta$, where $d_z$ is the dimension of the random vector $Z$, $\|\cdot\|_2$ denotes the $l_2$ norm, $0 < \beta < \infty$, and $I_a(z)$, called the generalized curvature matrix, is defined with entries

$$I_a(z)_{ij} = E\left( -\frac{\partial^2 \log p(X|Z=\widetilde{z})}{\partial \widetilde{z}_i \partial \widetilde{z}_j}\bigg|_{\widetilde{z}=a} \bigg| Z = z \right) = \left( \frac{\partial^2}{\partial \widetilde{z}_i \partial \widetilde{z}_j} \int \log \frac{p(x|z)}{p(x|\widetilde{z})} p(x|z) dx \right)\bigg|_{\widetilde{z}=a}.$$

**Assumption 2 (Smoothness on $p(z)$).** Assume that the probability density function $p(z)$ is twice continuously differentiable. Let $H_p(z)$ denote the Hessian matrix of $p(z)$ with respect to $z$. We assume that $\|H_p(z)\|_2 \le c_{d_z}$ holds almost everywhere, where $c_{d_z}$ depends only on the dimension $d_z$ of the random vector $Z$.

Note that Assumption 1 can be viewed as an extension of the requirement that the maximum eigenvalue of the Fisher information matrix for $x$ with respect to $z$ is bounded. Assumptions 1 and 2 can be validated when $(X, Z)$ follows a multivariate Gaussian distribution (MVD) and when $Z$ follows a MVD, respectively.

For $Z$, we define $Z_n^{(l)}$ as its $l$-nearest neighbor for $l = 1, ..., k$. According to Algorithm 3, each $Z_n^{(l)}$ is selected with probability $1/k$. Let $\xi$ be a random variable with a probability mass function $P(\xi = l) = 1/k$ for $l = 1, ..., k$, and $\xi$ is independent of both $Y$ and $Z$. Given $Z$, we denote by $\widetilde{X}$ the sample generated by the $k$-nearest-neighbor local sampling mechanism, which follows the distribution $\widehat{p}(x|Z) = p(x|Z_n^{(1)})^{1\{\xi=1\}} \times \ldots \times p(x|Z_n^{(k)})^{1\{\xi=k\}}$. The sketch proof of Theorem 2 proceeds as follows. We first apply Pinsker's inequality, which relates the total variation distance between $p(x|Z)$ and $\widehat{p}(x|Z)$ to their KL divergence. We then establish a connection between the KL divergence and the discrepancy between $Z$ and $Z_n^{(l)}$. Finally, by Lemma 1, we derive Theorem 2.

**Theorem 2.** *Under Assumptions 1 and 2, we have $d_{TV}\{p(x|Z), \widehat{p}(x|Z)\} = o_p(1)$ as $n \to \infty$.*

**Remark 1.** *The constant $\beta$ in Assumption 1 is used to establish an upper bound on the KL divergence between $p(x|Z)$ and $p(x|Z_n^{(l)})$, which can be used to bound $d_{TV}\{p(x|Z), \widehat{p}(x|Z)\}$ through Pinsker's inequality.*

In Theorem 3, we bound the excess type I error conditionally on $Y$ and $Z$ by the total variation distance between $\widehat{p}(\cdot|Z)$ and $p(\cdot|Z)$.

**Theorem 3.** *Assume $H_0 : X \perp\!\!\!\perp Y|Z$ is true. Under Assumptions 1 and 2, for any significance level $\alpha \in (0, 1)$, the $p$-value obtained from Algorithm 3 satisfies $P(p \le \alpha|Y, Z) \le \alpha + d_{TV}\{p(\cdot|Z), \widehat{p}(\cdot|Z)\}$.*

The type I error rate can be unconditionally controlled based on Theorems 2 and 3, which implies that $P(p \le \alpha|H_0) \le \alpha + o(1)$ as $n$ approaches infinity. Now we turn to analyze the power of our test in asymptotic scenarios in Theorem 4. Some intuitive explanations on both type I error rate and power can be found in the third paragraph of Section 3.2.

**Theorem 4.** *For any $\alpha \in (0,1)$ and the number of repetitions $B_n$ satisfying $B_n \to \infty$, let $p$ be the p-value calculated in Algorithm 3. Under Assumptions 1 and 2 and the Assumptions stated in the Supplementary Materials, $P(p \le \alpha | H_1) \to 1$ as $n \to \infty$.*

**Remark 2.** *The condition $B_n \to \infty$ is mild and has been made in [7, 43]. The consistency of our test heavily relies on the consistency of the CMI estimator, which hinges on whether the joint density of $(X, Y', Z)$ generated by the 1-NN sampling (Algorithm 1), denoted as $\phi$, approximates $p(x,z)p(y|z)$ (denoted as $g$) well, in terms of TV distance. $\beta$ and $c_{d_z}$ in Assumptions 1 and 2 are used to bound $d_{TV}(\phi, g)$.*

**Remark 3.** *According to Theorem 4, our test can achieve consistency against all the alternatives stated in $H_1$. In contrast, the method proposed by [43] only achieves consistency against a subset of the alternatives in $H_1$, and [30] does not provide any theoretical results on the testing power.*

## 5    Empirical Results

In this section, we present a comparative evaluation of our proposed method against the state-of-the-art (SOTA) methods on synthetic datasets. Our code is publicly available at: https://github.com/LeeShuai-kenwitch/NNLSCIT. We specifically compare our method with six commonly used competitive CI testing methods: KCIT [47], LPCIT[40], CCIT [41], CMIknn [38], GCIT [5], and NNSCIT [30]. We set the number of repetitions $B = 200$ and the neighbor order $k = 7$ for our tests. The XGBoost classifier was used in all of our experiments. Further elaboration on the choice of $k$ is given in Figure 3 in the Supplementary Materials. We set the significance level to $\alpha = 0.05$ and report the type I error rate and the testing power under $H_1$ for all methods evaluated in our experiments. All the results are presented as an average over 200 independent trials. We provide additional simulation studies and real data analysis in the Supplementary Materials.

**Scenario I: the post-nonlinear model.** The datasets used in our experiments are generated using the post-nonlinear model similar to those in [47, 16, 5, 40]. Specifically, we define $(X, Y, Z)$ under $H_0$ and $H_1$ as follows:

$$H_0 : X = f_1(\overline{Z} + 0.5\epsilon_x), \ Y = f_2(\overline{Z} + 0.5\epsilon_y),$$
$$H_1 : X = f_1(\overline{Z} + 0.5\epsilon_x) + 0.5\epsilon_b, Y = f_2(\overline{Z} + 0.5\epsilon_y) + 0.5\epsilon_b, \quad (9)$$

where $\overline{Z}$ represents the sample mean of $Z = (z_1, \ldots, z_{d_z})$, all $z_l$ in $Z$, $\epsilon_x, \epsilon_y$ and $\epsilon_b$ are i.i.d. samples generated from the standard Gaussian or the standard Laplace distribution, and $f_1$ and $f_2$ are randomly sampled from the set $\{x, x^2, x^3, \tanh(x), \cos(x)\}$.

We conduct a comparative analysis of various tests from two perspectives. The results are shown in Figure 1. First, we fix the sample size at $n = 500$ and vary the dimension of $Z$ from 30 to 100. In both the Gaussian and Laplace cases, our test shows good and stable performance in controlling type I error and achieving high power under $H_1$ as the dimension of $Z$ increases. LPCIT and NNSCIT have satisfactory performance in controlling type I error, but LPCIT loses power under $H_1$ when the dimension exceeds 70, and NNSCIT has inadequate power for all dimensions. Although GCIT, CCIT, and KCIT have adequate power under $H_1$, they inflate the type I error in almost all scenarios. CMIknn shows weak performance on both type I error and testing power. Figure 6 in the Supplementary Materials contains additional results on the performance of various tests in low dimensions of $Z$ ranging from 5 to 30, further demonstrating the superiority of our test. Second, we vary the sample size from 300 to 2000 while fixing the dimension of $Z$ at 80. Our test maintains good control of the type I error and achieves high power, with the power approaching 1 when the sample size exceeds 500. However, LPCIT, NNSCIT, and CMIknn lose power under $H_1$, while GCIT, CCIT, and KCIT either always or sometimes fail to control the type I error well.

In the Supplementary Materials, Figure 7 shows the timing performance of all methods for a single test. Our test is found to be highly computationally efficient even when dealing with large sample sizes and high-dimensional conditioning sets. In contrast, CMIknn and CCIT for sample sizes exceeding 1000, and LPCIT for dimension of $Z$ higher than 50 are impractical due to their prohibitively long running time.

**Scenario II: the heavy tailed model.** We compare the performance of our test with the SOTA tests under the heavy tailed error distributions, as described in [10]. The data $(X, Y, Z)$ is generated

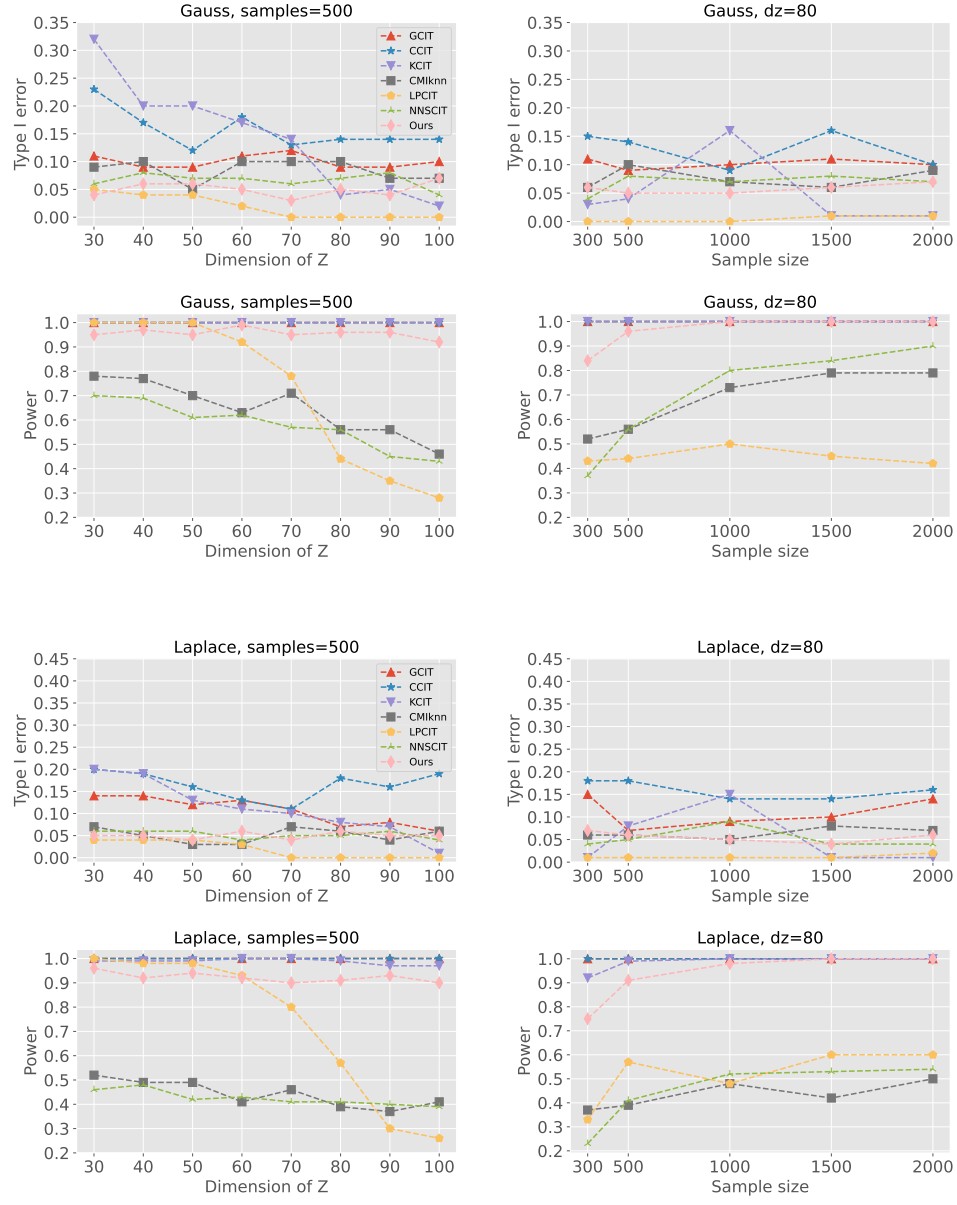

Figure 1: Comparison of the type I error (lower is better) and power (higher is better) of our method with six SOTA methods on the post-nonlinear model under Gaussian or Laplace distributions in Scenario I. **Left:** The results when varying the dimension of $Z$. **Right:** The results when varying the sample size.

according to the following model:

$$H_0 : X = \overline{Z} + \epsilon_1, \, Y = \overline{Z} + \epsilon_2,$$
$$H_1 : X = \overline{Z} + \epsilon_1, Y = \overline{Z} + \epsilon_1 + \epsilon_2, \quad (10)$$

where $\overline{Z}$ is the sample mean of $Z = (z_1, \ldots, z_{d_z})$, $z_l$'s are i.i.d. samples generated from the standard Gaussian distribution, and $\epsilon_1$ and $\epsilon_2$ are independently generated from the standard Cauchy distribution. We keep the sample size fixed at $n = 500$.

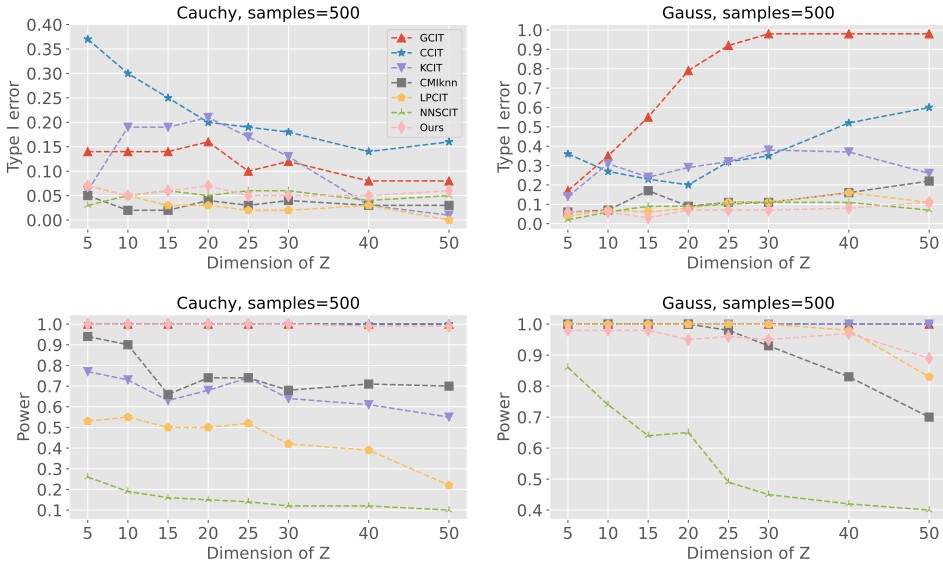

Figure 2: Comparison of the type I error (lower is better) and power (higher is better) of our method with six SOTA methods in Scenarios II and III. **Left:** The results in Scenario II. **Right:** The results in Scenario III.

**Scenario III: the chain structure.** We further use the chain structure $Y \to Z \to X$ in [30] to validate the proposed test. The detailed procedure for generating the simulated data can be found in the Supplementary Materials. The sample size is fixed at $n = 500$.

Figure 2 demonstrates that our test effectively controls the type I error and obtains adequate power in both Scenarios II and III under various dimensions of $Z$.

## 6 Conclusion

In this paper, we introduce a novel method for conducting the conditional independence testing. Theoretical analysis shows that our test is asymptotically valid and consistent against all alternatives in $H_1$. Extensive experiments on both synthetic and real datasets demonstrate that our method consistently outperforms commonly used SOTA methods. Moreover, our approach maintains computational efficiency even when the sample size and/or dimension of the conditioning set increase. Our method has the potential to enhance the applicability of causal discovery to real-world problems, such as gene regulatory networks or complex social networks, and to facilitate the discovery of relationships and patterns in complex systems. Ethically, we believe that our rather fundamental work has minimal potential for misuse.

## Acknowledgments

Dr. Ziqi Chen's work was partially supported by the National Key R&D Program of China (2021YFA1000100 and 2021YFA1000101), National Natural Science Foundation of China (NSFC) (12271167, 72331005 and 11871477), Natural Science Foundation of Shanghai (21ZR1418800) and Basic Research Project of Shanghai Science and Technology Commission (22JC1400800). Dr. Christina Dan Wang's work was partially supported by the National Natural Science Foundation of China (NSFC) (12271363 and 11901395). We thank the anonymous reviewers for their helpful comments.

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

# Supplementary Materials for "$K$-Nearest-Neighbor Local Sampling Based Conditional Independence Testing"

## A   Theoretical results

### A.1   Proof of Lemma 1

*Proof.* Recall that $Z$ is a random vector taking values in Euclidean space $(\mathbb{R}^{d_z}, \|\cdot\|_2)$, where $d_z$ is the dimension of $Z$ and $\|\cdot\|_2$ is Euclidean distance. $Z_1, Z_2, \ldots, Z_n$ are i.i.d. random vectors according to $p(z)$. For a fixed $z \in \mathbb{R}^{d_z}$, we denote by $Z_n^{(1)}(z), \ldots, Z_n^{(n)}(z)$ a reordering of $Z_1, Z_2, \ldots, Z_n$ according to the increasing values of $\|Z_i - z\|_2$, that is,

$$\|Z_n^{(1)}(z) - z\|_2 \le \ldots \le \|Z_n^{(n)}(z) - z\|_2.$$

Define the set $G = \{z \in \mathbb{R}^{d_z} \mid \forall \delta > 0, \ P(\{\omega : Z(\omega) \in S_z(\delta)\}) > 0\}$, where $S_z(\delta) = \{x \in \mathbb{R}^{d_z} \mid \|x - z\|_2 \le \delta\}$. For convenience, we omit $\omega$ in probability in the following paper. For example, write $P(Z \in S_z(\delta))$ instead of $P(\{\omega : Z(\omega) \in S_z(\delta)\})$. By definition, for $z \in G$, $\forall \delta > 0$, $P(Z \in S_z(\delta)) > 0$. Let $G^c$ be the complement of $G$. Then, for $z \in G^c$, $\exists r_z > 0$, s.t. $\forall r < r_z$, $P(Z \in S_z(r)) = 0$. Note that $P(\|Z_n^{(k)}(z) - z\|_2 > \delta) = P(Z_n^{(k)}(z) \notin S_z(\delta))$.

In order to prove $\|Z_n^{(k)}(Z) - Z\|_2 \to 0$ a.s., it is sufficient to prove $\forall \delta > 0$, $\lim_{n\to\infty} P(\sup_{m\ge n}\|Z_m^{(k)}(Z) - Z\|_2 > \delta) = 0$. We can obtain

$$P(\sup_{m\ge n}\|Z_m^{(k)}(Z) - Z\|_2 > \delta) \le P(\{\sup_{m\ge n}\|Z_m^{(k)}(Z) - Z\|_2 > \delta\} \cap \{Z \in G\}) + P(Z \in G^c)$$

$$= \int_G P(\sup_{m\ge n}\|Z_m^{(k)}(z) - z\|_2 > \delta)p(z)dz + P(Z \in G^c)$$

$$\le \int_G \sum_{m\ge n} P(Z_m^{(k)}(z) \notin S_z(\delta))p(z)dz + P(Z \in G^c). \quad (11)$$

First, consider the first term of (11). We have

$$
\begin{aligned}
P(Z_m^{(k)}(z) \notin S_z(\delta)) = {}& P(Z_m^{(1)}(z), Z_m^{(2)}(z), \ldots, Z_m^{(m)}(z) \notin S_z(\delta)) \\
& + P(Z_m^{(1)}(z) \in S_z(\delta), Z_m^{(2)}(z), \ldots, Z_m^{(m)}(z) \notin S_z(\delta)) \\
& + P(Z_m^{(1)}(z), Z_m^{(2)}(z) \in S_z(\delta), Z_m^{(3)}(z), \ldots, Z_m^{(m)}(z) \notin S_z(\delta)) \\
& + \ldots + P(Z_m^{(1)}(z), \ldots, Z_m^{(k-1)}(z) \in S_z(\delta), Z_m^{(k)}(z), \ldots, Z_m^{(m)}(z) \notin S_z(\delta)).
\end{aligned}
$$

By setting $P(Z \in S_z(\delta)) = \gamma$, we have

$$
\begin{aligned}
P(Z_m^{(k)}(z) \notin S_z(\delta)) = {}& (1-\gamma)^m + C_m^1\gamma(1-\gamma)^{m-1} + C_m^2\gamma^2(1-\gamma)^{m-2} + \ldots \\
& + C_m^{k-1}\gamma^{k-1}(1-\gamma)^{m-k+1}. \quad (12)
\end{aligned}
$$

Consider the $j$-th term of (12). Let $C_1 := \gamma^j/j!$, $C_2 := C_1 e^j$ and $C_3 := C_2 e^j(1-\gamma)^{-j}$. By using Stirling's approximation, we have

$$
\begin{aligned}
\lim_{m\to\infty} C_m^j \gamma^j (1-\gamma)^{m-j} &= \lim_{m\to\infty} \frac{m!}{(m-j)!j!}\gamma^j(1-\gamma)^{m-j} \\
&= \lim_{m\to\infty} C_1 \frac{\sqrt{2\pi m}(\frac{m}{e})^m}{\sqrt{2\pi(m-j)}(\frac{m-j}{e})^{m-j}}(1-\gamma)^{m-j} \\
&= \lim_{m\to\infty} C_2 \sqrt{\frac{m}{(m-j)}}\frac{m^m}{(m-j)^{m-j}}(1-\gamma)^{m-j} \\
&= \lim_{m\to\infty} C_2 e^j(m+j)^j(1-\gamma)^m \\
&= \lim_{m\to\infty} C_2 e^j m^j(1-\gamma)^{m-j} \\
&= \lim_{m\to\infty} C_3 m^j(1-\gamma)^m.
\end{aligned}
$$

Thus, there exists $C_4 > 0$ such that $P(Z_m^{(k)}(z) \notin S_z(\delta)) \le C_4 m^{k-1}(1-\gamma)^m$ when $m$ is large enough. It holds that

$$\frac{C_4 m^{k-1}(1-\gamma)^m}{m^{-2}} = C_4 m^{k+1}(1-\gamma)^m \to 0, \quad \text{as} \quad m \to +\infty.$$

Thus, for $z \in G$ and $n$ large enough, $\forall \delta > 0$, we have $P(Z_m^{(k)}(z) \notin S_z(\delta)) = o(m^{-2})$ and

$$\sum\nolimits_{m \ge n} P(Z_m^{(k)}(z) \notin S_z(\delta)) \le \sum\nolimits_{m \ge n} \frac{1}{m^2},$$

which shows $\lim_{n \to \infty} \sum_{m \ge n} P(Z_m^{(k)}(z) \notin S_z(\delta)) = 0$ for $z \in G$. So by Lebesgue dominated convergence theorem, we obtain $\lim_{n \to \infty} \int_G \sum_{m \ge n} P(Z_m^{(k)}(z) \notin S_z(\delta)) p(z) dz = 0$.

Second, we consider the second term of (11). To prove that $P(Z \in G^c) = 0$, we aim to construct a countable open cover of $G^c$ and show that the probability of the random vector $Z$ falling into each of these open balls is zero. By the property of $G^c$, for every $z \in G^c$, there exists $r_z > 0$ such that for all $r < r_z$, $P(Z \in S_z(r)) = 0$. Furthermore, using the separability of Euclidean space and the density of the rational number set, we can approximate $z$ using points from $\mathbb{Q}^{d_z}$ with $\mathbb{Q}$ being the rational number set. Therefore, for every $z \in G^c$, there exist $x \in \mathbb{Q}^{d_z} \cap S_z(\frac{r_z}{3})$ and $r \in \mathbb{Q} \cap (\frac{r_z}{3}, \frac{2r_z}{3})$, such that $z \in S_x(r) \subseteq S_z(r_z)$. Because $P(Z \in S_z(r_z)) = 0$, we conclude that $P(Z \in S_x(r)) = 0$. Define

$$\mathcal{F} := \{S_x(r) | \exists z \in G^c, \text{ such that } z \in S_x(r) \subseteq S_z(r_z) \text{ with } x \in \mathbb{Q}^{d_z} \text{ and } r \in \mathbb{Q}\}.$$

Note that the elements in set $\mathcal{F}$ are mutually distinct. By the construction of $S_x(r)$, $\mathcal{F}$ forms a countable open cover of $G^c$, and the probability of $Z$ falling into each open ball in $\mathcal{F}$ is zero. Using the monotonicity and countable additivity properties of probability, we have $P(Z \in G^c) \le P(Z \in \cup_{S_x(r) \in \mathcal{F}} S_x(r)) \le \sum_{S_x(r) \in \mathcal{F}} P(Z \in S_x(r)) = 0$. Thus, we conclude that $P(Z \in G^c) = 0$.

We therefore conclude that, $\forall \delta > 0$, $\lim_{n \to \infty} P(\sup_{m \ge n} \|Z_m^{(k)}(Z) - Z\|_2 > \delta) = 0$. This finish the proof.

## A.2 Proof of Theorem 2

*Proof.* By Pinsker's inequality, we have

$$d_{TV}\{p(x|Z), \widehat{p}(x|Z)\} \le \sqrt{D_{KL}\{p(x|Z), \widehat{p}(x|Z)\}/2}.$$

Note that $I\{\xi = 1\} + \ldots + I\{\xi = k\} = 1$. By the definition of $\widehat{p}(x|Z)$, we obtain

$$
\begin{aligned}
D_{KL}\{p(x|Z), \widehat{p}(x|Z)\} &= \int p(x|Z) \log\left\{\frac{p(x|Z)}{p(x|Z_n^{(1)})^{I\{\xi=1\}} \times \ldots \times p(x|Z_n^{(k)})^{I\{\xi=k\}}}\right\} dx \\
&= \int p(x|Z) \log \prod_{l=1}^{k} \frac{p(x|Z)^{I\{\xi=l\}}}{p(x|Z_n^{(l)})^{I\{\xi=l\}}} dx \\
&= \sum_{l=1}^{k} I\{\xi = l\} \int p(x|Z) \log \frac{p(x|Z)}{p(x|Z_n^{(l)})} dx \\
&= \sum_{l=1}^{k} I\{\xi = l\} D_{KL}\{p(x|Z) \| p(x|Z_n^{(l)})\}.
\end{aligned}
$$

Then, by Taylor's expansion, we have

$$
\begin{aligned}
D_{KL}&\{p(x|Z) \| p(x|Z_n^{(l)})\} \\
&= D_{KL}\{p(x|Z) \| p(x|Z)\} + \frac{\partial}{\partial z'} D_{KL}\{p(x|Z) \| p(x|z')\}\Big|_{z'=Z} (Z_n^{(l)} - Z) \\
&\quad + \frac{1}{2}(Z_n^{(l)} - Z)^T \frac{\partial^2}{\partial z' \partial z'^T} D_{KL}\{p(x|Z) \| p(x|z')\}\Big|_{z'=a} (Z_n^{(l)} - Z),
\end{aligned}
$$

where $a = \lambda Z + (1 - \lambda)Z_n^{(l)}$ with $0 \leq \lambda \leq 1$.

Note that $D_{KL}\{p(x|Z)||p(x|Z)\} = \int p(x|Z) \log \frac{p(x|Z)}{p(x|Z)} dx = 0$. By Lemma 1 and Assumptions 1 and 2, we have

$$\frac{\partial}{\partial z'} D_{KL}\{p(x|Z)||p(x|z')\}\Big|_{z'=Z} = -\int p(x|Z) \cdot \frac{\partial}{\partial z'} \log p(x|z')\Big|_{z'=Z} dx$$

$$= -\frac{\partial}{\partial z'} \int p(x|z')dx\Big|_{z'=Z} = 0$$

and

$$\frac{\partial^2}{\partial z'\partial z'^T} D_{KL}\{p(x|Z)||p(x|z')\}\Big|_{z'=a} = -\int p(x|Z) \cdot \frac{\partial^2}{\partial z'\partial z'^T} \log p(x|z')\Big|_{z'=a} dx = I_a(Z).$$

This means

$$D_{KL}\{p(x|Z)||p(x|Z_n^{(l)})\} = \frac{1}{2}(Z_n^{(l)} - Z)^T I_a(Z)(Z_n^{(l)} - Z).$$

Note that $Z_n^{(l)} \to Z$ a.s. implies that $Z_n^{(l)}$ converges to $Z$ in probability. Then $\forall \delta > 0$, for $\epsilon$ defined in Assumption 1, we have

$$P(D_{KL}\{p(x|Z)||p(x|Z_n^{(l)})\} > \delta) \leq P(\{D_{KL}\{p(x|Z)||p(x|Z_n^{(l)})\} > \delta\} \cap \{\|Z_n^{(l)} - Z\|_2 \leq \epsilon\})$$

$$+ P(\|Z_n^{(l)} - Z\|_2 > \epsilon)$$

$$\leq P\left(\frac{1}{2}\beta\|Z_n^{(l)} - Z\|_2^2 > \delta\right) + P(\|Z_n^{(l)} - Z\|_2 > \epsilon)$$

$$= o(1).$$

Thus, $D_{KL}\{p(x|Z)||p(x|Z_n^{(l)})\} = o_p(1)$.

Because $I\{\xi = l\} \leq 1$ for $l = 1, 2, ..., k$, and $k$ is finite, we obtain

$$D_{KL}\{p(x|Z), \widehat{p}(x|Z)\} = \sum_{l=1}^{k} I\{\xi = l\} D_{KL}\{p(x|Z)||p(x|Z_n^{(l)})\} = o_p(1).$$

Finally, we conclude that $d_{TV}\{p(x|Z), \widehat{p}(x|Z)\} \leq \sqrt{D_{KL}\{p(x|Z), \widehat{p}(x|Z)\}/2} = o_p(1)$.

### A.3  Proof of Theorem 3

To prove Theorem 3, the following two lemmas are needed.

**Lemma 5.** *Let $\acute{X}$ be drawn from $\widehat{p}(\cdot|Z)$, independently of $Y$. $\acute{X}^{(1)}, \ldots, \acute{X}^{(B)}$ are i.i.d. samples drawn from the k-nearest-neighbor local sampling mechanism based on $(\acute{X}, Y, Z)$. For any statistic $T$, the $B + 1$ statistics*

$$T(\acute{X}, Y, Z), T(\acute{X}^{(1)}, Y, Z), \ldots, T(\acute{X}^{(B)}, Y, Z)$$

*are exchangeable conditionally on $Y$ and $Z$.*

*Proof.* We have that the $B+1$ triples $(\acute{X}, Y, Z), (\acute{X}^{(1)}, Y, Z), \ldots, (\acute{X}^{(B)}, Y, Z)$ are i.i.d. samples drawn from the same mechanism after conditionally on $\acute{X}_{()}, Y$ and $Z$, where $\acute{X}_{()}$ is the order statistic of $\acute{X}$. Note that, $T$ is measurable. Thus, $T(\acute{X}, Y, Z), T(\acute{X}^{(1)}, Y, Z), \ldots, T(\acute{X}^{(B)}, Y, Z)$ are i.i.d. after conditionally on $\acute{X}_{()}, Y$ and $Z$. Conditionally on $\acute{X}_{()}, Y$ and $Z$, denote their cumulative conditional distribution as $F(\cdot|\acute{X}_{()}, Y, Z)$. Denote $\acute{X}$ as $\acute{X}^{(0)}$. Then, for any $t_0, \ldots, t_B \in \mathbb{R}$ and

any permutation $\pi = (\pi_{(0)}, \ldots, \pi_{(B)})$ of the indices $\{0, 1, \ldots, B\}$, we have

$$
\begin{aligned}
&P(T(\acute{\boldsymbol{X}}^{(0)}, \boldsymbol{Y}, \boldsymbol{Z}) \leq t_0, T(\acute{\boldsymbol{X}}^{(1)}, \boldsymbol{Y}, \boldsymbol{Z}) \leq t_1, \ldots, T(\acute{\boldsymbol{X}}^{(B)}, \boldsymbol{Y}, \boldsymbol{Z}) \leq t_B | \boldsymbol{Y}, \boldsymbol{Z}) \\
&= E_{\acute{\boldsymbol{X}}_{()}|\boldsymbol{Y}, \boldsymbol{Z}}\{P(T(\acute{\boldsymbol{X}}^{(0)}, \boldsymbol{Y}, \boldsymbol{Z}) \leq t_0, T(\acute{\boldsymbol{X}}^{(1)}, \boldsymbol{Y}, \boldsymbol{Z}) \leq t_1, \ldots, T(\acute{\boldsymbol{X}}^{(B)}, \boldsymbol{Y}, \boldsymbol{Z}) \leq t_B | \acute{\boldsymbol{X}}_{()}, \boldsymbol{Y}, \boldsymbol{Z})\} \\
&= E_{\acute{\boldsymbol{X}}_{()}|\boldsymbol{Y}, \boldsymbol{Z}}\{P(T(\acute{\boldsymbol{X}}^{(0)}, \boldsymbol{Y}, \boldsymbol{Z}) \leq t_0 | \acute{\boldsymbol{X}}_{()}, \boldsymbol{Y}, \boldsymbol{Z}), \ldots, P(T(\acute{\boldsymbol{X}}^{(B)}, \boldsymbol{Y}, \boldsymbol{Z}) \leq t_B | \acute{\boldsymbol{X}}_{()}, \boldsymbol{Y}, \boldsymbol{Z})\} \\
&= E_{\acute{\boldsymbol{X}}_{()}|\boldsymbol{Y}, \boldsymbol{Z}}\left\{ \prod_{i=0}^{B} F(t_i | \acute{\boldsymbol{X}}_{()}, \boldsymbol{Y}, \boldsymbol{Z}) \right\} \\
&= E_{\acute{\boldsymbol{X}}_{()}|\boldsymbol{Y}, \boldsymbol{Z}}\{P(T(\acute{\boldsymbol{X}}^{(\pi_{(0)})}, \boldsymbol{Y}, \boldsymbol{Z}) \leq t_0, \ldots, T(\acute{\boldsymbol{X}}^{(\pi_{(B)})}, \boldsymbol{Y}, \boldsymbol{Z}) \leq t_B | \acute{\boldsymbol{X}}_{()}, \boldsymbol{Y}, \boldsymbol{Z})\} \\
&= P(T(\acute{\boldsymbol{X}}^{(\pi_{(0)})}, \boldsymbol{Y}, \boldsymbol{Z}) \leq t_0, \ldots, T(\acute{\boldsymbol{X}}^{(\pi_{(B)})}, \boldsymbol{Y}, \boldsymbol{Z}) \leq t_B | \boldsymbol{Y}, \boldsymbol{Z}).
\end{aligned}
$$

Thus, the desired result follows.

Let $\overset{d}{=}$ denotes equality in distribution. We present the following Lemma:

**Lemma 6.** *For any two bi-tuples $(\boldsymbol{U}, \boldsymbol{V})$ and $(\boldsymbol{U}', \boldsymbol{V}')$, if $\forall \boldsymbol{u}$, $(\boldsymbol{V}|\boldsymbol{U} = \boldsymbol{u}) \overset{\mathrm{d}}{=} (\boldsymbol{V}'|\boldsymbol{U}' = \boldsymbol{u})$, we have $d_{TV}\{(\boldsymbol{U}, \boldsymbol{V}), (\boldsymbol{U}', \boldsymbol{V}')\} = d_{TV}(\boldsymbol{U}, \boldsymbol{U}')$.*

*Proof.* Denote the joint density functions of $(\boldsymbol{U}, \boldsymbol{V})$ and $(\boldsymbol{U}', \boldsymbol{V}')$ by $p_{\boldsymbol{U}, \boldsymbol{V}}(\boldsymbol{u}, \boldsymbol{v})$ and $p_{\boldsymbol{U}', \boldsymbol{V}'}(\boldsymbol{u}', \boldsymbol{v}')$, respectively. According to the equivalent definition of the TV distance, we obtain

$$
\begin{aligned}
d_{TV}\{(\boldsymbol{U}, \boldsymbol{V}), (\boldsymbol{U}', \boldsymbol{V}')\} &= \frac{1}{2} \iint \left| p_{\boldsymbol{U}, \boldsymbol{V}}(\boldsymbol{u}, \boldsymbol{v}) - p_{\boldsymbol{U}', \boldsymbol{V}'}(\boldsymbol{u}, \boldsymbol{v}) \right| d\boldsymbol{u} d\boldsymbol{v} \\
&= \frac{1}{2} \iint \left| p_{\boldsymbol{V}|\boldsymbol{U}}(\boldsymbol{v}|\boldsymbol{u}) p_{\boldsymbol{U}}(\boldsymbol{u}) - p_{\boldsymbol{V}'|\boldsymbol{U}'}(\boldsymbol{v}|\boldsymbol{u}) p_{\boldsymbol{U}'}(\boldsymbol{u}) \right| d\boldsymbol{u} d\boldsymbol{v} \\
&= \frac{1}{2} \iint p_{\boldsymbol{V}|\boldsymbol{U}}(\boldsymbol{v}|\boldsymbol{u}) \left| p_{\boldsymbol{U}}(\boldsymbol{u}) - p_{\boldsymbol{U}'}(\boldsymbol{u}) \right| d\boldsymbol{u} d\boldsymbol{v} \\
&= \frac{1}{2} \int \left[ \int p_{\boldsymbol{V}|\boldsymbol{U}}(\boldsymbol{v}|\boldsymbol{u}) d\boldsymbol{v} \right] \left| p_{\boldsymbol{U}}(\boldsymbol{u}) - p_{\boldsymbol{U}'}(\boldsymbol{u}) \right| d\boldsymbol{u} \\
&= \frac{1}{2} \int \left| p_{\boldsymbol{U}}(\boldsymbol{u}) - p_{\boldsymbol{U}'}(\boldsymbol{u}) \right| d\boldsymbol{u} \\
&= d_{TV}(\boldsymbol{U}, \boldsymbol{U}').
\end{aligned}
$$

Now we present the proof of Theorem 3:

*Proof.* Let $\widetilde{\boldsymbol{X}}^{(1)}, \ldots, \widetilde{\boldsymbol{X}}^{(B)}$ be i.i.d. drawn from the $k$-nearest-neighbor local sampling mechanism, see Algorithm 3. Now let $\acute{\boldsymbol{X}}$ be an additional sample drawn from $\widehat{p}(\cdot|\boldsymbol{Z})$ independently of $\boldsymbol{Y}$. Let $\acute{\boldsymbol{X}}^{(1)}, \ldots, \acute{\boldsymbol{X}}^{(B)}$ be i.i.d. drawn from the $k$-nearest-neighbor local sampling mechanism after we observe $\acute{\boldsymbol{X}}$ instead of $\boldsymbol{X}$. Because $(\acute{\boldsymbol{X}}^{(1)}, \ldots, \acute{\boldsymbol{X}}^{(B)})$, conditionally on $\acute{\boldsymbol{X}}, \boldsymbol{Y}$ and $\boldsymbol{Z}$, is generated from the same mechanism as $(\widetilde{\boldsymbol{X}}^{(1)}, \ldots, \widetilde{\boldsymbol{X}}^{(B)})$, conditionally on $\boldsymbol{X}, \boldsymbol{Y}$ and $\boldsymbol{Z}$, for all $\boldsymbol{x} \in \mathbb{R}^n$, we have

$$
((\widetilde{\boldsymbol{X}}^{(1)}, \ldots, \widetilde{\boldsymbol{X}}^{(B)})|\boldsymbol{X} = \boldsymbol{x}, \boldsymbol{Y}, \boldsymbol{Z} \overset{\mathrm{d}}{=} ((\acute{\boldsymbol{X}}^{(1)}, \ldots, \acute{\boldsymbol{X}}^{(B)})|\acute{\boldsymbol{X}} = \boldsymbol{x}, \boldsymbol{Y}, \boldsymbol{Z}).
$$

Then, by applying Lemma 6, we obtain

$$
\begin{aligned}
&d_{TV}\{(\boldsymbol{X}, \widetilde{\boldsymbol{X}}^{(1)}, \ldots, \widetilde{\boldsymbol{X}}^{(B)}|\boldsymbol{Y}, \boldsymbol{Z}), (\acute{\boldsymbol{X}}, \acute{\boldsymbol{X}}^{(1)}, \ldots, \acute{\boldsymbol{X}}^{(B)}|\boldsymbol{Y}, \boldsymbol{Z})\} \\
&= d_{TV}\{(\boldsymbol{X}|\boldsymbol{Y}, \boldsymbol{Z}), (\acute{\boldsymbol{X}}|\boldsymbol{Y}, \boldsymbol{Z})\} = d_{TV}\{p(\cdot|\boldsymbol{Z}), \widehat{p}(\cdot|\boldsymbol{Z})\}.
\end{aligned}
$$

Define $\chi_\alpha^B := \left\{ (\boldsymbol{x}, \boldsymbol{x}^{(1)}, \ldots, \boldsymbol{x}^{(B)}) \middle| \left[ 1 + \sum_{b=1}^{B} 1\{T(\boldsymbol{x}^{(b)}, \boldsymbol{Y}, \boldsymbol{Z}) \geq T(\boldsymbol{x}, \boldsymbol{Y}, \boldsymbol{Z})\} \right] / (1 + B) \leq \alpha \right\}$, where $1(\cdot)$ is the indicator function. Note that in our case, the statistic $T$ is selected to be $\widehat{\mathrm{CMI}}$. Then,

it follows that

$$P(p \leq \alpha | \mathbf{Y}, \mathbf{Z}) = P((\mathbf{X}, \widetilde{\mathbf{X}}^{(1)}, \ldots, \widetilde{\mathbf{X}}^{(B)}) \in \chi_\alpha^B | \mathbf{Y}, \mathbf{Z})$$

$$= P((\acute{\mathbf{X}}, \acute{\mathbf{X}}^{(1)}, \ldots, \acute{\mathbf{X}}^{(B)}) \in \chi_\alpha^B | \mathbf{Y}, \mathbf{Z}) + P((\mathbf{X}, \widetilde{\mathbf{X}}^{(1)}, \ldots, \widetilde{\mathbf{X}}^{(B)}) \in \chi_\alpha^B | \mathbf{Y}, \mathbf{Z})$$

$$- P((\acute{\mathbf{X}}, \acute{\mathbf{X}}^{(1)}, \ldots, \acute{\mathbf{X}}^{(B)}) \in \chi_\alpha^B | \mathbf{Y}, \mathbf{Z})$$

$$\leq P((\acute{\mathbf{X}}, \acute{\mathbf{X}}^{(1)}, \ldots, \acute{\mathbf{X}}^{(B)}) \in \chi_\alpha^B | \mathbf{Y}, \mathbf{Z})$$

$$+ d_{TV}\{(\mathbf{X}, \widetilde{\mathbf{X}}^{(1)}, \ldots, \widetilde{\mathbf{X}}^{(B)} | \mathbf{Y}, \mathbf{Z}), (\acute{\mathbf{X}}, \acute{\mathbf{X}}^{(1)}, \ldots, \acute{\mathbf{X}}^{(B)} | \mathbf{Y}, \mathbf{Z})\}$$

$$= P((\acute{\mathbf{X}}, \acute{\mathbf{X}}^{(1)}, \ldots, \acute{\mathbf{X}}^{(B)}) \in \chi_\alpha^B | \mathbf{Y}, \mathbf{Z}) + d_{TV}\{p(\cdot | \mathbf{Z}), \widehat{p}(\cdot | \mathbf{Z})\}.$$

Applying Lemma 5 and the property of rank test, we obtain $P((\acute{\mathbf{X}}, \acute{\mathbf{X}}^{(1)}, \ldots, \acute{\mathbf{X}}^{(B)}) \in \chi_\alpha^B | \mathbf{Y}, \mathbf{Z}) \leq \alpha$. Finally, we have $P(p \leq \alpha | \mathbf{Y}, \mathbf{Z}) \leq \alpha + d_{TV}\{p(\cdot | \mathbf{Z}), \widehat{p}(\cdot | \mathbf{Z})\}$.

Because the TV distance is bounded by 1, marginalizing the above inequality over $\mathbf{Y}$ and $\mathbf{Z}$ and applying Theorem 2 and Lebesgue dominated convergence theorem lead to

$$P(p \leq \alpha | H_0) \leq \alpha + E(d_{TV}\{p(\cdot | \mathbf{Z}), \widehat{p}(\cdot | \mathbf{Z})\}) = \alpha + o(1).$$

### A.4 Proof of Theorem 4

Here we present the assumptions given in [34] that ensure the consistency of CMI estimator. We denote the point $(x, y, z)$ as $\omega \in \mathbb{R}^{d_x} \times \mathbb{R}^{d_y} \times \mathbb{R}^{d_z}$. Let $f(\omega) = p(x, y, z)$ be the joint density function of $(X, Y, Z)$, $g(\omega) = p(x, z)p(y|z)$ be the joint density function of $(X, Y, Z)$ under $H_0$, and $\phi(\omega) = \phi(x, y, z)$ be the joint density of $(X, Y', Z)$ produced by Algorithm 1. For a given point $\omega$, $\gamma(\omega) = P(l = 1 | \omega)$ represents the predicted positive label probability generated by a classifier. When the prediction is from a classifier with parameter $\theta$, it is denoted as $\gamma_\theta(\omega)$. For convenience, the parameter $\theta$ in $\gamma$ is dropped when it is understood from the context. According to Algorithm 2, $P(l = 1) = P(l = 0) = 1/2$. Define the population binary-cross entropy loss over the joint distribution of data and label as $\text{BCE}(\gamma) = -(E_{Wl}(l \log \gamma(W) + (1 - l) \log(1 - \gamma(W)))$.

**Assumption (A1)**: $f(\cdot)$ and $\phi(\cdot)$ admit densities in a compact subset $\mathcal{W} \subset \mathbb{R}^{d_x} \times \mathbb{R}^{d_y} \times \mathbb{R}^{d_z}$.

**Assumption (A2)**: For some constant $\alpha, \beta > 0$, $\alpha \leq f(\omega), \phi(\omega) \leq \beta, \forall \omega$.

**Assumption (A3)**: Clip predictions such that $\gamma(\omega) \in [\tau, 1 - \tau], \forall \omega$, with $0 < \tau \leq \alpha/(\alpha + \beta)$.

**Assumption (A4)**: The classifier class $\mathcal{C}_\theta$ is parametrized by $\theta$ within a compact domain $\Theta \subset \mathbb{R}^h$. There exists a constant $K$ such that $\|\theta\|_2 \leq K$, and the classifier's output is $L$-Lipschitz with respect to $\theta$.

**Assumption (A5)**: $\int p(z)^{1-1/d} dz \leq C_5, \forall d \geq 2$, where $C_5$ is a constant.

We denote the CMI estimator $\widehat{\text{CMI}}$ based on Algorithm 2 by $\widehat{D}_{KL}^{(n)}(f \| \phi)$. The true CMI of $(X, Y, Z)$ is $\text{CMI} = I(X; Y | Z) = D_{KL}(f \| g)$. Then we have the following Lemma:

**Lemma 7.** *Under Assumptions 1 and 2 and (A1)-(A5), we have $\widehat{D}_{KL}^{(n)}(f \| \phi) \xrightarrow{P} D_{KL}(f \| g)$.*

*Proof.* By the definition of convergence in probability, it is sufficient to prove $\forall \delta > 0, \forall \eta > 0, \exists N$, when $n > N$, $P(|\widehat{D}_{KL}^{(n)}(f \| \phi) - D_{KL}(f \| g)| > \delta) < \eta$.

Note that

$$P(|\widehat{D}_{KL}^{(n)}(f \| \phi) - D_{KL}(f \| g)| > \delta)$$

$$= P(|\widehat{D}_{KL}^{(n)}(f \| \phi) - D_{KL}(f \| \phi) + D_{KL}(f \| \phi) - D_{KL}(f \| g)| > \delta).$$

Applying Theorem 1 in [34], we have $\widehat{D}_{KL}^{(n)}(f \| \phi) - D_{KL}(f \| \phi) \xrightarrow{P} 0$, which means for $\delta/2 > 0$ and $\eta > 0, \exists N_1$, when $n > N_1$, $P(|\widehat{D}_{KL}^{(n)}(f \| \phi) - D_{KL}(f \| \phi)| > \delta/2) < \eta$.

Now consider the term $D_{KL}(f \| \phi) - D_{KL}(f \| g)$. Let $\gamma'$ be the point-wise minimizer of binary-cross entropy loss based on $f(\omega)$ and $\phi(\omega)$, and $\gamma''$ be the point-wise minimizer of binary-cross entropy

loss based on $f(\omega)$ and $g(\omega)$. Applying Lemma 3 in [34], we have $\gamma'(\omega)/\{1-\gamma'(\omega)\} = f(\omega)/\phi(\omega)$ and $\gamma''(\omega)/\{1-\gamma''(\omega)\} = f(\omega)/g(\omega)$. Then, by the definition of KL divergence, it follows that

$$
\begin{aligned}
|D_{KL}(f||\phi) - D_{KL}(f||g)| &= \left| E_{f(\omega)} \log \frac{f(\omega)}{\phi(\omega)} - E_{f(\omega)} \log \frac{f(\omega)}{g(\omega)} \right| \\
&= \left| E_{f(\omega)} \left( \log \frac{\gamma'(\omega)}{1-\gamma'(\omega)} - \log \frac{\gamma''(\omega)}{1-\gamma''(\omega)} \right) \right| \\
&\leq E_{f(\omega)} \left| \log \frac{1-\gamma'(\omega)}{\gamma'(\omega)} - \log \frac{1-\gamma''(\omega)}{\gamma''(\omega)} \right| \\
&\leq \frac{1-\tau}{\tau} E_{f(\omega)} \left| \frac{1-\gamma'(\omega)}{\gamma'(\omega)} - \frac{1-\gamma''(\omega)}{\gamma''(\omega)} \right| \\
&= \frac{1-\tau}{\tau} E_{f(\omega)} \left| \frac{\phi(\omega) - g(\omega)}{f(\omega)} \right| \\
&= \frac{1-\tau}{\tau} \iiint |\phi(x,y,z) - g(x,y,z)| dx dy dz \\
&= \frac{2(1-\tau)}{\tau} d_{TV}(\phi, g).
\end{aligned}
$$

The second inequality follows from Lagrange's mean value theorem and Assumption (A3).

Applying Theorem 1 in [41], $\forall \epsilon_1 \leq \epsilon$ with $\epsilon$ being defined in Assumption 1, we have $d_{TV}(\phi, g) \leq b(n)$, where

$$
b(n) = \frac{1}{2} \sqrt{ \frac{\beta}{4} \frac{C_5 2^{1/d_z} \Gamma(1/d_z)}{(n\gamma_{d_z})^{1/d_z} d_z} + \frac{\beta \epsilon_1 G(2c_{d_z} \epsilon_1^2)}{4} } + \exp\left( -\frac{1}{2} n\gamma_{d_z} c_{d_z} \epsilon_1^{d_z+2} \right) + G(2c_{d_z} \epsilon_1^2).
$$

Here, $\beta$ is defined in Assumption 1, $C_5$ is defined in Assumption (A5), $d_z$ is the dimension of $Z$, $\Gamma(\cdot)$ is the gamma function, $\gamma_{d_z}$ is the volume of the unit radius $l_2$ ball in $\mathbb{R}^{d_z}$, $c_{d_z}$ is defined in Assumption 2, and $\forall \delta > 0$, $G(\delta) = P(p(Z) \leq \delta)$.

Because $\epsilon_1$ can be arbitrary small, we conclude that $\lim_{n\to\infty} b(n) = 0$. So we arrive at $\lim_{n\to\infty} |D_{KL}(f||\phi) - D_{KL}(f||g)| = 0$, which means for $\delta/2 > 0$, $\exists N_2$, when $n > N_2$, $|D_{KL}(f||\phi) - D_{KL}(f||g)| < \delta/2$.

Then for $\delta > 0$ and $\eta > 0$, take $N = \max(N_1, N_2)$, when $n > N$,

$$
\begin{aligned}
& P(|\widehat{D}_{KL}^{(n)}(f||\phi) - D_{KL}(f||\phi) + D_{KL}(f||\phi) - D_{KL}(f||g)| > \delta) \\
& \leq P(|\widehat{D}_{KL}^{(n)}(f||\phi) - D_{KL}(f||\phi)| + |D_{KL}(f||\phi) - D_{KL}(f||g)| > \delta) \\
& \leq P(|\widehat{D}_{KL}^{(n)}(f||\phi) - D_{KL}(f||\phi)| > \delta/2) < \eta
\end{aligned}
$$

holds. This finish the proof.

We therefore conclude that $\widehat{\text{CMI}}$ is a consistent estimator of CMI. When considering $\widehat{\text{CMI}}^{(b)}$ based on the sample $(\widetilde{\boldsymbol{X}}^{(b)}, \boldsymbol{Y}, \boldsymbol{Z})$ ($b = 1, \ldots, B$) drawn from the $k$-nearest-neighbor local sampling mechanism as depicted in Algorithm 3, we can state: Under Assumptions 1, 2, (A1)-(A2) with $f(\omega)$ and $\phi(\omega)$ replaced by densities of the distribution of $(\widetilde{X}, Y, Z)$ and the corresponding 1-NN distribution, respectively, and Assumptions (A3)-(A5), $\forall b = 1, \ldots, B$, $\widehat{\text{CMI}}^{(b)}$ is a consistent estimator of $\text{CMI}^{(b)}$, where $\text{CMI}^{(b)} = I(\widetilde{X}^{(b)}; Y|Z)$. Now let's present the proof of Theorem 4.

*Proof.* Write $P(\cdot|H_1)$ as $P_{H_1}(\cdot)$. By Markov inequality, it follows that

$$P_{H_1}(p > \alpha) = P_{H_1}\left(\frac{1 + \Sigma_{b=1}^{B_n} 1(\widehat{\mathrm{CMI}}^{(b)} \geq \widehat{\mathrm{CMI}})}{1 + B_n} > \alpha\right)$$

$$\leq \frac{1}{\alpha(1 + B_n)} E_{H_1}\left(1 + \sum_{b=1}^{B_n} 1(\widehat{\mathrm{CMI}}^{(b)} \geq \widehat{\mathrm{CMI}})\right)$$

$$= \frac{1}{\alpha(1 + B_n)} + \frac{B_n}{\alpha(1 + B_n)} P_{H_1}\left(\widehat{\mathrm{CMI}}^{(1)} \geq \widehat{\mathrm{CMI}}\right)$$

$$\leq \frac{1}{\alpha(1 + B_n)} + \frac{1}{\alpha} P_{H_1}\left(\widehat{\mathrm{CMI}}^{(1)} \geq \widehat{\mathrm{CMI}}\right).$$

Because $\widetilde{X}^{(1)} \perp\!\!\!\perp Y|Z$, $\mathrm{CMI}^{(1)} = I(\widetilde{X}^{(1)}; Y|Z) = 0$. Then, $\forall \delta > 0$,

$$P_{H_1}\left(\widehat{\mathrm{CMI}}^{(1)} \geq \widehat{\mathrm{CMI}}\right) \leq P_{H_1}\left(\{\widehat{\mathrm{CMI}} \leq \widehat{\mathrm{CMI}}^{(1)}\} \cap \{|\widehat{\mathrm{CMI}}^{(1)} - \mathrm{CMI}^{(1)}| \leq \delta\}\right)$$

$$+ P_{H_1}\left(|\widehat{\mathrm{CMI}}^{(1)} - \mathrm{CMI}^{(1)}| > \delta\right)$$

$$\leq P_{H_1}(\widehat{\mathrm{CMI}} \leq \delta) + P_{H_1}\left(|\widehat{\mathrm{CMI}}^{(1)} - \mathrm{CMI}^{(1)}| > \delta\right).$$

Next, we have

$$P_{H_1}(\widehat{\mathrm{CMI}} \leq \delta) \leq P_{H_1}(\{\widehat{\mathrm{CMI}} \leq \delta\} \cap \{|\widehat{\mathrm{CMI}} - \mathrm{CMI}| \leq \delta\}) + P_{H_1}(|\widehat{\mathrm{CMI}} - \mathrm{CMI}| > \delta)$$

$$\leq P_{H_1}(\mathrm{CMI} - \delta \leq \widehat{\mathrm{CMI}} \leq \delta) + P_{H_1}(|\widehat{\mathrm{CMI}} - \mathrm{CMI}| > \delta).$$

Thus, we conclude that

$$P_{H_1}(p \leq \alpha) \geq 1 - \frac{1}{\alpha(1 + B_n)} - \frac{1}{\alpha}\left[P_{H_1}(\mathrm{CMI} - \delta \leq \widehat{\mathrm{CMI}} \leq \delta)\right.$$

$$\left. + P_{H_1}(|\widehat{\mathrm{CMI}} - \mathrm{CMI}| > \delta) + P_{H_1}\left(|\widehat{\mathrm{CMI}}^{(1)} - \mathrm{CMI}^{(1)}| > \delta\right)\right].$$

Under $H_1$, $\mathrm{CMI} > 0$. Take $\delta = \mathrm{CMI}/4 > 0$, we obtain

$$\lim_{n \to \infty} P_{H_1}(p \leq \alpha) \to 1.$$

# B  Additional Empirical Results

## B.1  The choice of the neighbor order $k$

To investigate the impact of the parameter $k$ on our proposed approach, we employ a linear uniform model. To accomplish this, we generate synthetic data in the following manner:

$$H_0 : X = \epsilon_x, \ Y = \epsilon_y, \ \text{and} \ Z \sim \text{Uniform}(-1, 1),$$
$$H_1 : X = \epsilon_x, \ Y = \alpha X + 0.5\epsilon_y, \ \text{and} \ Z \sim \text{Uniform}(-1, 1), \tag{13}$$

where $\epsilon_x$ and $\epsilon_y$ are generated independently from the uniform distribution over the interval $[-1, 1]$. The parameter $\alpha$ is randomly generated within the range of $[0, 2]$. As is shown in Figure 3, our method achieves effective control of type I error and exhibits the highest power under $H_1$ across all dimensions when $k = 7$. Therefore, we consistently set $k = 7$ in all experiments.

## B.2  Empirical results for Scenario (13)

We demonstrate the effectiveness of our approach and compare it with alternative methods in Scenario (13). The results are shown in Figure 4, which pertains to high-dimensional $Z$, and Figure 5, which focuses on low-dimensional $Z$. The results consistently demonstrate that our test achieves favorable performance in terms of the type I error and power under $H_1$. Although LPCIT, CMIknn, and NNSCIT effectively control the type I error, they exhibit noticeably lower power compared to our method, when the dimension exceeds 60, often by a substantial margin. Furthermore, KCIT, GCIT, and CCIT all yield high power under $H_1$, but they either always or sometimes suffer from inflated type I errors.

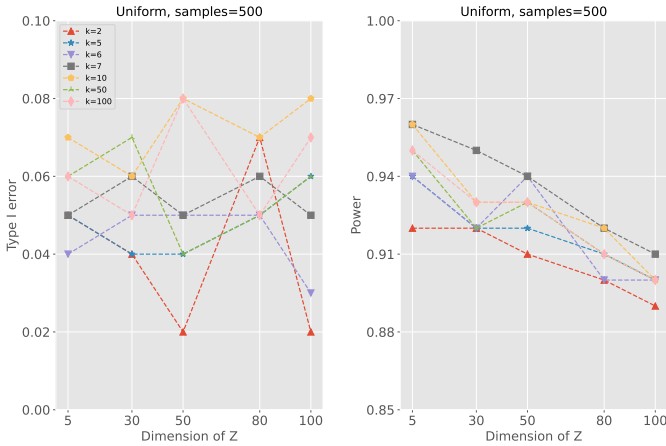

Figure 3: Comparison of the type I error (lower is better) and power under $H_1$ (higher is better) for our test in Scenario (13) across different values of $k$.

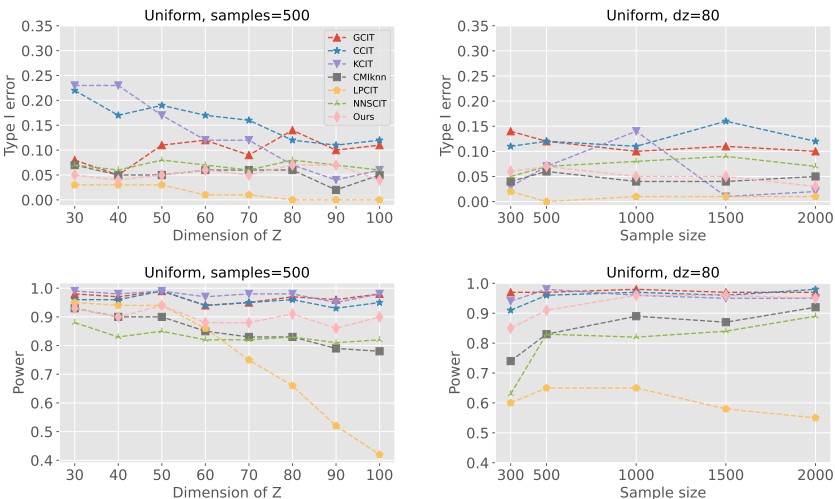

Figure 4: Comparison of the type I error (lower is better) and power under $H_1$ (higher is better) of our test with six SOTA tests in Scenario (13). **Left:** The results when varying the dimension of $Z$. **Right:** The results when varying the sample size.

### B.3 Additional empirical results for Scenario I

In Figure 6, we present the type I error and power under $H_1$ in low dimensions of $Z$ ranging from 5 to 30 for Scenario I (Eq. (9)) with Gaussian or Laplace noises. It can be observed that our test and LPCIT consistently achieve good and stable performance in terms of type I error and power under $H_1$, when the dimensionality of $Z$ is lower than 30. On the other hand, GCIT, CCIT, and KCIT exhibit high power under $H_1$ but fail to control the type I error. NNSCIT and CMIknn demonstrate relatively good control of type I errors but lack sufficient power under $H_1$.

### B.4 Computational efficiency analysis

Figure 7 shows the timing performance of all methods for a single test under Scenario I with Laplace noises. Our test is found to be highly computationally efficient even when dealing with large

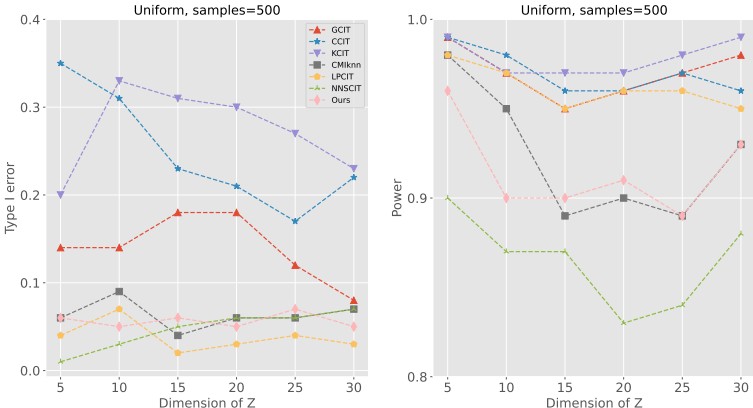

Figure 5: Comparison of the type I error (lower is better) and power under $H_1$ (higher is better) of our test with six SOTA tests in Scenario (13).

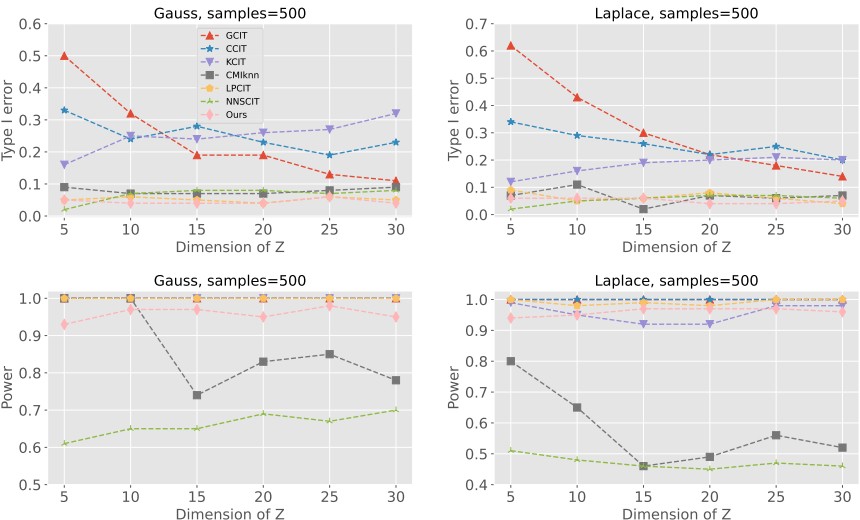

Figure 6: Comparison of the type I error (lower is better) and power under $H_1$ (higher is better) of our method with six SOTA methods on the post-nonlinear model under Gaussian or Laplace distributions in Scenario I. **Left:** The results under Gaussian distribution. **Right:** The results under Laplace distribution.

sample sizes and high-dimensional conditioning sets. In contrast, CMIknn and CCIT for sample sizes exceeding 1000, and LPCIT for dimension of Z higher than 50 are impractical due to their prohibitively long running time.

### B.5 The detailed experimental setup for Scenario III

For the chain structure $Y \to Z \to X$, we generate synthetic data as follows:

$$H_0 : Y \sim N(1,1), \ Z = Ya + \epsilon_1, \ X = Z^T b + \epsilon_2,$$
$$H_1 : Y \sim N(1,1), \ Z = Ya + \epsilon_1, \ X = Z^T b + Y + \epsilon_2,$$

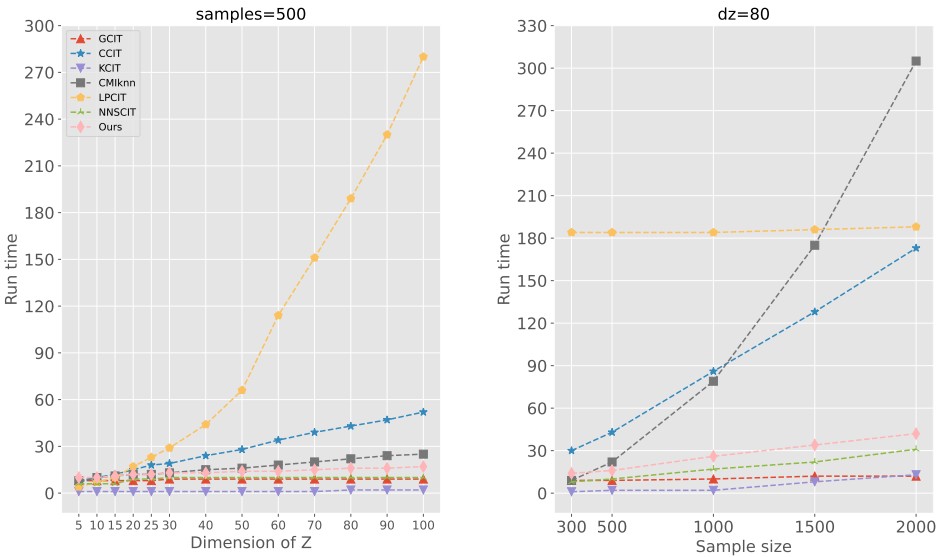

Figure 7: Running times in seconds as a function of sample size or dimension of $Z$ on the post-nonlinear model under Laplace distribution in Scenario I. **Left:** The results when varying the dimension of $Z$. **Right:** The results when varying the sample size.

where $a$ and $b$ are both $d_z$-dimensional, the entries of $a$ and $b$ are both randomly and uniformly sampled from $[0, 0.3]$, $\epsilon_1$ is generated from a $d_z$-dimensional standard multivariate Gaussian distribution, and $\epsilon_2$ is sampled from a standard Gaussian distribution.

## C   Real Data Analysis

In order to showcase the superior performance of our test, we conduct a comparative evaluation against other state-of-the-art (SOTA) CI tests using real datasets. We assess the effectiveness of our method along with six SOTA approaches on two specific datasets: the ABALONE dataset [9] and the Flow-Cytometry dataset [39].

### C.1   Real ABALONE dataset

The ABALONE dataset [9] comprises measurements obtained from a study conducted to predict the age of abalones based on their physical characteristics. The dataset is publicly available at the UCI Machine Learning Repository and can be downloaded from https://archive.ics.uci.edu/ml/datasets/abalone. In our evaluation, we consider the graph structure recovered by [31] as the ground truth, as depicted in Figure 4 of their paper. This graph represents the causal relationships among the 8 variables in the dataset. We specifically select 35 CI relations and 35 non-CI relations from this graph. The philosophy used is that a node $X$ is independent of all other nodes $Y$ in the graph when conditioned on its parents, children, and parents of children [26, 41]. Additionally, if there exists a direct edge between node $X$ and node $Y$ in the graph, they are never conditionally independent given any other set of variables. As a result, the conditioning set $Z$ can be arbitrarily selected from the remaining nodes. The dataset consists of 4177 samples, and $d_z$ varies from 1 to 6.

In order to evaluate the performance of various tests, we utilize precision, recall, and F-score as evaluation metrics. Precision is calculated as TP/(TP+FP), where TP represents the number of true CI instances correctly identified, and FP represents the number of non-CI instances incorrectly identified as CI. Recall is calculated as TP/(TP+FN), where FN represents the number of CI instances not identified. The F-score is then computed as the harmonic mean of precision and recall, given by $2 \times$ precision $\times$ recall / (precision + recall) [29]. TN represents the number of correctly identified true

non-CI instances. Table 1 presents the results for all methods. It should be noted that we do not record the results for GCIT as it does not correctly identify any CI relations. Our approach successfully identifies 31 CI relations and 32 non-CI relations, achieving the highest F-score among the testing methods, while maintaining high precision and recall.

Table 1: The TP, TN, precision (pre), recall (rec) and F-score of our test and six SOTA methods for the real ABALONE dataset.

| Method | TP | TN | Pre | Rec | F-score |
|---|---|---|---|---|---|
| KCIT | 5 | 35 | 1 | 0.1429 | 0.2501 |
| CCIT | 12 | 34 | 0.9231 | 0.3429 | 0.5 |
| CMIknn | 22 | 35 | 1 | 0.6286 | 0.7720 |
| LPCIT | 5 | 35 | 1 | 0.1429 | 0.2501 |
| NNSCIT | 33 | 6 | 0.5323 | 0.9429 | 0.6805 |
| Ours | 31 | 32 | 0.9118 | 0.8857 | 0.8986 |

**C.2 Real Flow-Cytometry dataset**

The Flow-Cytometry dataset is a widely used benchmark in the field of causal structure learning [36, 48]. This dataset captures the expression levels of proteins and phospholipids in human cells [39]. The data can be obtained from the website https://www.science.org/doi/10.1126/science.1105809. In our evaluation, we consider the consensus graph proposed in [33] as the ground truth, which has also been adopted by [41] for verifying CI relations. Figure 5(a) in [33] illustrates the causal relationships among the 11 proteins in the dataset. Following the philosophy outlined in Section C.1, we select 50 CI relations and 40 non-CI relations from this graph. The number of samples is 1755 and $d_z$ varies from 1 to 9.

Table 2 presents the results for all tests. Our method outperforms other approaches by correctly identifying 47 CI relations and achieving the highest recall and F-score.

Table 2: The TP, TN, precision (pre), recall (rec) and F-score of our test and six SOTA methods for the real Flow-Cytometry dataset.

| Method | TP | TN | Pre | Rec | F-score |
|---|---|---|---|---|---|
| KCIT | 32 | 30 | 0.7619 | 0.64 | 0.6957 |
| CCIT | 33 | 29 | 0.75 | 0.66 | 0.7021 |
| CMIknn | 41 | 26 | 0.7455 | 0.82 | 0.7810 |
| GCIT | 40 | 24 | 0.7143 | 0.8 | 0.7547 |
| LPCIT | 38 | 25 | 0.7170 | 0.76 | 0.7379 |
| NNSCIT | 33 | 26 | 0.7021 | 0.66 | 0.6804 |
| Ours | 47 | 23 | 0.7344 | 0.94 | 0.8246 |

