# OpenReview forum: "K-Nearest-Neighbor Local Sampling Based Conditional Independence Testing"
_NeurIPS.cc/2023/Conference — NeurIPS 2023 poster_

### Official Review · Reviewer_P7Sn · 2023-07-05

**Soundness:** 3 good
**Presentation:** 4 excellent
**Contribution:** 3 good
**Rating:** 7
**Confidence:** 3

**Summary:**

This paper argues that conditional independence (CI) testing becomes challenging because of high-dimensional conditioning variables and limited data samples. To address these issues, the authors propose a testing approach incorporating a class-based conditional mutual information (CMI) and a $k$-nearest-neighbor local sampling strategy. Theoretical analysis demonstrates its asymptotic control of type I error and consistency against all alternative hypotheses. Extensive empirical results on synthetic and real data show that the proposed method achieves computational efficiency, decent performance under different scenarios, and robustness towards heavy-tailed data.

**Strengths:**

This idea is intrigued by two significant challenges of CI testing, i.e., high-dimensional conditioning variables and limited data samples.
The authors do a good job of motivating this work, and the proposed test approach is simple but effective.
Extensive empirical results show that the proposed method achieves a better trade-off between type I error rate and testing power.
In addition, the paper is very well written and easy to understand.

**Weaknesses:**

My main doubts/concerns regarding the paper are the following:
- Line 104-105 claim that the $k$-nearest-neighbor local sampling strategy is an alternative to the binning strategy. However, this paper does not provide a comparison between the proposed testing approach and the methods employing a binning strategy [1,2].
- In Figure 3, the results on synthetic data show that the performance is sensitive to the selection of $k$. Therefore, it is unreasonable to apply $k=7$ in all experiments, especially in real data.

Minor:
- Empirical analysis shows that the proposed method achieves a better trade-off between type I error rate and testing power. It is hard to say that the proposed method outperforms existing SOTA methods.

[1] I. Kim, et al., "Local Permutation Tests for Conditional Independence", The Annals of Statistic 2022.

[2] D. Margaritis, "Distribution-free Learning of Bayesian Network Structure in Continuous Domains", AAAI 2005.

**Questions:**

1. Can the authors add more discussions on the difference between the $k$-nearest-neighbor local sampling strategy and the binning strategy? What are the advantages of the $k$-nearest-neighbor local sampling strategy?
2. Can the authors provide the optimal $k$ on real data?

**Limitations:**

Yes, the authors have addressed the societal impact of their work.

---

> ### Author Rebuttal · Authors · 2023-08-06
>
> We thank the reviewer for the helpful review!
>
> **Weakness 1**. The binning strategy can only handle conditioning variables $Z$  with very few dimensions. Please see the response for Question 1. We therefore did not consider the comparison with the methods employing a binning strategy in the paper.
>
> **Weakness 2**.  Thank you for this important question. Conditional independence testing is an unsupervised problem, since the ground truth information regarding conditional independence or non-conditional independence is unavailable. Therefore, the cross-validation method does not work for determining the optimal $k$ in practical scenarios. The task of selecting the optimal $k$ appears to be quite challenging in the context of CIT. We will delve into developing methods to identify the optimal parameter $k$ in our future research. Nevertheless, utilizing $k=7$ in the experiments and the two real data analyses yields very good performances,  even though it may not represent the optimal choice for $k$. It is noted that,  in the works related to CIT, such as Runge (2018), Bellot and van der Schaar (2019), and Scetbon et al. (2022), the hyperparameters are chosen using synthetic data and then applied to other experiments and real data analyses.
>
> **Weakness (Minor)**: When assessing the effectiveness of a test, it's crucial that it showcases a higher statistical power compared to other methods, while simultaneously effectively controlling the type I error. On one hand, if a test fails to control the type I error well, it cannot be considered as reliable. On the other hand, when a test successfully controls the type I error, its power is then compared to that of other tests. If it exhibits superior power, it can be confidently regarded as more favorable and superior. Based on this perspective, it is observed from the experiments that our method outperforms the commonly used competitive SOTA methods. For this statement, we will replace “existing SOTA methods” with “the commonly used competitive SOTA methods”.
>
> **Question 1**. The binning strategy can only handle conditioning variables $Z$  with very few dimensions. Kim et al. (2022) proposed two binning strategies. The first binning strategy is to partition the conditional variable $Z$ into $M$ bins of equal size. Then, we permute the X-samples in each bin randomly and perform local permutation tests. The second binning strategy is to partition $Z$ into $M$ coarser bins and further partition each coarser bin into fine bins in which permutation occurred. Kim et al. (2022) used both strategies to process univariate $Z$ in all of the experiments. Margaritis (2005) adopted a recursive-median binning strategy and  can not handle the high-dimensional conditional variable $Z$ either. In fact, we have conducted simulation studies and confirmed that the binning strategy can only handle the condition variable $Z$ with very  few dimensions.
>
> We give explanations for the limitations of the binning strategy in addressing the high-dimensional conditional variable $Z$.
> Binning serves as a technique to partition samples that exhibit notable similarity with respect to variable $Z$ into distinct bins. Shuffling $X$ within each bin maintains the relationship between $X$ and $Z$, as opposed to directly permuting $X$ throughout the entire dataset. Nevertheless, employing the binning method in high-dimensional scenarios poses challenges (Berrett et al., 2020). One challenge stems from the intricate task of selecting bins where the samples within each bin exhibit notable similarity concerning the variable $Z$. Another challenge arises from the generation of an excessive number of bins in high-dimensional $Z$. For instance, consider a scenario where the dimension of $Z$ is $100$. By applying the first binning strategy described in Kim et al. (2022) to partition each dimension of $Z$, we would generate an excessive number of bins for $Z$. Specifically, the number of bins is $M^{100}$. Even with a small value of $M=2$, the binary binning technique would produce a staggering $2^{100}$ bins. As highlighted by Kim et al. (2022), redundant bins can result in a reduction of test power. More specifically, a substantial number of bins within the high-dimensional space of Z consist of just one or two samples. On one hand, as per the definition of $U$-statistics in the computation of test statistics in Kim et al. (2022), only bins comprising a minimum of $4$ samples are included in the computation of $U$-statistics. Consequently, those bins containing just one or two samples do not contribute to the calculation of the test statistics. This leads to an underutilization of samples in these bins. On the other hand, significant similarity among various permutations can arise when numerous bins consist of merely one or two samples. This similarity can result in insufficient statistical power, and may also undermine effective control of the type I error rate.
>
> Different from the binning strategy involving permutation, the k-nearest-neighbor local sampling strategy serves as an alternative sampling approach. This approach equips us with pseudo samples $\widetilde{\pmb{X}}^{(b)}$ ($b=1,\cdots,B$) that exhibit considerable dissimilarity while closely approximating the true conditional distribution in terms of the total variation distance, as evidenced by our Lemma 1. Especially in scenarios with high-dimensional $Z$, the k-nearest-neighbor local sampling strategy, compared to the permutation-based binning approach, effectively resolves concerns like maintaining the relationship between $X$ and $Z$ for pseudo sample $\tilde{X}$, as well as addressing issues of sample underutilization and pronounced similarity among permutations. This results in adequate statistical power and good control of the type I error rate within the context of high-dimensional $Z$.
>
> **Question 2**. Please see our response to Weakness 2.

---

> > ### Comment · Reviewer_P7Sn · 2023-08-15
> > **Official Comment by Reviewer P7Sn**
> >
> > Dear authors, thanks for the clarifications. The comments and references from the authors solve my primary concerns, and I've updated my score.

---

> > > ### Author Response · Authors · 2023-08-15
> > > **Thank Reviewer P7Sn**
> > >
> > > Dear Reviewer P7Sn,
> > >
> > > Thanks for your positive feedback and raising the score.
> > >
> > > Authors

---

### Official Review · Reviewer_671y · 2023-07-06

**Soundness:** 4 excellent
**Presentation:** 4 excellent
**Contribution:** 4 excellent
**Rating:** 7
**Confidence:** 4

**Summary:**

In this paper, the authors present a method for conditional independence (CI) testing based on a conditional mutual information (CMI) estimator.
This CMI estimator is based on a classifier over 1-NN samples. From the CMI estimates, the authors present a hypothesis testing procedure for CI based on multiple iterations over estimates of CMI on K-NN samples.

The authors present a theoretical analysis of their approach and the empirical results show that their approach has low Type 1 error and high power. The empirical section shows also that their approach remains consistent as the number of dimensions increase for the conditioning variables.

**Strengths:**

The paper is well written and technically sound. The contribution builds on previous approaches but is an original method for the important task of CI.
The method is very promising given the theoretical and empirical results providing balanced detection of TP and TN links in real-world datasets as shown by the top F-scores.
The use of a single classifier for the CMI estimation is advantageous as the CI test is usually executed multiple times for graph detection.

**Weaknesses:**

I find the paper really compelling and useful in many other ML tasks. However, I believe the empirical evaluation could have gone a step further.
In particular, I believe a distillation procedure would be helpful to show which part of your approach contributes to the significant results. More precisely, it would be great to see how good your CMI estimator is. A comparison against two-classifier approaches and even a synthetic dataset where you have access to the ground truth conditional distributions would be very useful.
This could show how tight the bound is given your estimator. Furthermore, it would be nice to see the performance of your approach given different classifiers, e.g., DNNs or Random forest methods as mentioned in your paper.

In the presentation, I would make the figures larger.

**Questions:**

I might have missed it but it wasn't clear to me, which classifier you used in your experiments?

What do you mean by $V_0^{test}$ in line 7 of alg. 2? is that any instance over the union of test sets in f and g?

How long does your approach take compared to the others? After all, a large number of B repetitions could significantly add up.


**Limitations:**

I don't expect any potential negative societal impact from this work

---

> ### Author Rebuttal · Authors · 2023-08-07
>
> We thank the reviewer for the important review!
>
> **Weakness 1**.  To show which parts of our approach contribute to the significant results, we have done some additional experiments as per your suggestions. Specifically, the classifiers we evaluate include DNNs, XGBoost, and Random forest. The simulation dataset is generated according to 'Scenario I: The post-nonlinear model' in the paper. We maintain the distribution as the standard Gaussian, with the sample size $n=500$ and dimensions of $Z$ set to $10$ and $50$.
>
> First, we assess the effects of different classifiers on CMI estimation. Under the null hypothesis $H_0$, the true CMI value is zero. We calculate the mean squared errors (MSEs) of the CMI estimators using different classifiers under $H_0$. These MSE values are averaged over 100 independent trials. It is observed from Table 3 in the pdf file (attached in the global response) that the MSEs of the CMI estimators based on the three classifiers exhibit remarkable similarity. Furthermore, these three classifier-based CMI estimation methods all yield estimators that closely approximate the true value of $0$ under the null hypothesis.
>
> However, calculating the true value of CMI under $H_1$ is highly complex. Therefore, we do not compute the MSE in this case. Instead, we further utilize the thresholding approach to evaluate the performances. To be specific, we conduct the experiment independently 100 times under the assumption of $H_0$. While $H_0$ being true implies $I(X;Y|Z)=0$, the CMI estimator from a finite-sized dataset seldom equals zero due to statistical variation. We simply set the threshold at $0.1$ for the CMI estimator: if $\hat{I}>0.1$, we reject $H_0$, and we then count the instances of falsely rejecting $H_0$. Similarly, under $H_1$, we threshold the CMI estimator at $0.1$ and record the number of correct reject $H_0$. Table 4 in the pdf file (attached in the global response) records the results obtained. We observed that, XGBoost-CMI  has the highest empirical type I errors $0.12 (0.19)$ and  empirical testing power $0.85 (0.83)$, and DNNs-CMI  has the lowest  empirical type I error $0.05 (0.07)$ and  empirical testing power $0.38 (0.12)$. In practice,  choosing an appropriate threshold value presents a significant challenge due to the inherent difficulty in accurately estimating the statistical variance of the CMI estimator. Therefore, relying on a threshold-based scheme for testing conditional independence is considered unwise. To enhance the effectiveness of CI testing based on the classifier-CMI estimator,  we propose in this paper applying the k-nearest-neighbor local sampling strategy to approximate the conditional distribution $X|Z$ that encodes the null hypothesis. This scheme takes into account the statistical variation inherent in the CMI estimation. It is not only simple and effective, but also has theoretical guarantee (See Theorem 2).
>
> Second, we employ the same dataset utilized for CMI estimation to evaluate the performance of our CMI estimation method using various classifiers equipped with the k-nearest-neighbor local sampling approach. We present the type I error rate, testing power, and timing performance for a single test for each method in Table 5 in the pdf file (attached in the global response). These results are averaged over 100 independent trials. Upon examination of the results in Tables 4 and 5 in the pdf file (attached in the global response), we observe that when using the threshold value as the basis for testing conditional independence, the XGBoost-CMI method exhibits inflated type I error, while the DNNs-CMI method demonstrates low testing power. However, when incorporating the k-nearest-neighbor local sampling scheme, the performance of both methods improves. Specifically, the  empirical type I error rate of XGBoost-CMI decreases from $0.12 (0.17)$ to $0.06 (0.06)$, and its  empirical testing power increases from $0.85 (0.83)$ to $0.97 (0.96)$. Moreover, the  empirical testing power of DNNs-CMI  increases from $0.38 (0.12)$ to $0.85 (0.34)$. These results indicate that both a good classifier-based CMI estimator and the k-nearest-neighbor local sampling strategy are key factors contributing to the significant outcomes achieved by our test.
>
> **Weakness 2**. Thanks for your suggestion. We will make the figures larger in the final version.
>
> **Question 1**. The XGBoost classifier was used in all of our experiments.
>
> **Question 2**. After completing the classifier training in Algorithm 2 (line 7), we proceed to compute the prediction probability $P(l=1|w)$ for each feature $w$ in the test set $V^{test}$. The label of the sample $w$ (i.e., $l=1$ or $l=0$) is not required when computing $P(l=1|w)$. However, since $V^{test}$ contains labels for all samples, we introduce a new symbol $V_{0}^{test}$, which includes all the features in $V^{test}$ but excludes the labels.  $V_{0}^{test}$ is  essentially the union of the test sets in $f$ and $g$.
>
> **Question 3**. We report the timing performance of all methods for a single test in Figure 7 of the Supplementary Materials. Our test is found to be highly computationally efficient even when dealing with large sample sizes and high-dimensional conditioning sets. In contrast, CMIknn and CCIT for sample sizes exceeding 1000, and LPCIT for dimension of $Z$ higher than 50 are impractical due to their prohibitively long running time. For example, when $n=2000$ and $dz=80$, a single test of our method takes 42 seconds, GCIT 12 seconds, CCIT 173 seconds, KCIT 13 seconds, NNSCIT 31 seconds, CMIknn 305 seconds and LPCIT 188 seconds. In our approach, we set $B=200$.

---

> > ### Comment · Reviewer_671y · 2023-08-14
> >
> > I'm satisfied with the comments of the authors and I will keep my scoring as is.

---

> > > ### Author Response · Authors · 2023-08-15
> > > **Thank Reviewer 671y**
> > >
> > > Dear Reviewer 671y,
> > >
> > > Thanks for your positive feedback.
> > >
> > > Authors

---

### Official Review · Reviewer_xTib · 2023-07-07

**Soundness:** 3 good
**Presentation:** 3 good
**Contribution:** 3 good
**Rating:** 5
**Confidence:** 3

**Summary:**

In this paper, the authors propose a novel CMI estimator with a classifier-based approach. The estimated CMI is then used for conditional independence testing. This approach achieves asymptotic type I & II control and does not require prior assumptions on the distribution, types of correlations, or knowledge of the null distribution. Empirical results show promising performance.

**Strengths:**

1. This paper is well-written.
2. The proposed CMI is very natural for testing CI. The algorithm is easy to implement given any classifier.
3.  The proposed method is backed up by theory (Theorem 3 & 4).


**Weaknesses:**

1.  As authors have pointed out, prior works for estimating MI (e.g. Suzuki et al., 2008 and [3]) using density ratio do exist, and a conditional version of such an estimator is not exactly a huge leap forward (but no small feat either)

2. The control of type I & II is asymptotic, and it is unclear how useful they are when the sample size is small. The claim of Theorem 3 and 4 is unsurprising (due to the consistency of KNN and smoothness of conditional distribution). If the proof is non-trivial, please point out.

Suzuki et al., 2008 Approximating Mutual Information by Maximum Likelihood Density Ratio Estimation

**Questions:**

1. Why is the CMI estimated using classifiers rather than directly optimising (4) like f-GAN? Why not directly use the maximum of (4) to approximate the CMI? This way, you would not need to split the dataset into training and testing sets right?

**Limitations:**

Yes

---

> ### Author Rebuttal · Authors · 2023-08-06
>
> We thank the reviewer for the insightful review!
>
> **Weakness 1**. The Maximum Likelihood Mutual Information (MLMI) method proposed by Suzuki et al. (2008) directly models the density ratio and avoids density estimation. MLMI estimates the density ratio using maximum likelihood and formulates it as a convex optimization problem using Legendre-Fenchel duality. We may extend this to the conditional version. Nevertheless, the MLMI-based method heavily depends on selecting appropriate basis functions that can effectively capture the information in $Z$, and it can face challenges due to the curse of dimensionality (Mukherjee et al., 2020). In contrast, our approach employs the classifier-based estimation of the likelihood ratio, offering a solution capable of addressing high-dimensional challenges.
>
> **Weakness 2**. First, in the simulation study, we present  some results when the sample size is $n=300$ as can be seen in Figures 1 and 4. Even when the sample size is small, our proposed test  controls the type I error well and achieves adequate testing power. Note that, GCIT and CMIknn consider a minimum sample size of 500, and DGCIT, NNSCIT and CCIT all consider a minimum sample size of 1000.
>
> Second, we discuss the non-trivial points in the type-I error control of our method in Theorem 3. If the conditional distribution of $X|Z$ is known, the p-value is calculated by $$p:=\frac{1+\sum_{b=1}^{B}{1}(\widetilde{\mbox{CMI}}{}^{(b)}\geq \widehat{\mbox{CMI}})} {1+B},$$ as can be seen in Equation (8) in the paper. Because of the exchangeability of $B+1$ triples $(\pmb{X}, \pmb{Y}, \pmb{Z}),(\pmb{X}^{(1)}, \pmb{Y}, \pmb{Z}),\cdots,(\pmb{X}^{(B)}, \pmb{Y}, \pmb{Z})$ under $H_0$, this p-value is valid and  $P(p\leq \alpha |H_{0})\leq \alpha$ holds for any given $\alpha \in (0,1)$.  However, in our case, $X|Z$ is unknown. $\widetilde{\pmb{X}}$ is drawn from $\widehat{p}(\cdot|\pmb{Z})$ instead of $p(\cdot|\pmb{Z})$. Therefore, the existing argument based on the assumption of a known conditional distribution $X|Z$ does not apply in this context. So, in the proof of Theorem 3, we  introduce  an additional sample $\acute{\pmb{X}}$ drawn from $\widehat{p}(\cdot|\pmb{Z})$ independently of $\pmb{Y}$. We proceed to independently and identically draw $\acute{\pmb{X}}^{(1)}, \cdots, \acute{\pmb{X}}^{(B)}$ using the $k$-nearest neighbor local sampling mechanism based on $(\acute{\pmb{X}}, \pmb{Y}, \pmb{Z})$. Define $\chi\_{\alpha}^B:=\\{(\pmb{x},\pmb{x}^{(1)},\ldots , \pmb{x}^{(B)})\Big| \left[1+\sum\_{b=1}^B1(T(\pmb{x}^{(b)},\pmb{Y},\pmb{Z})\geq T(\pmb{x},\pmb{Y},\pmb{Z}))\right]\big/(1+B)\leq \alpha \\}$. The type I error of our test  conditionally  on $\pmb{Y}$ and $\pmb{Z}$ can be expressed as $P((\pmb{X},\widetilde{\pmb{X}}^{(1)},\ldots, \widetilde{\pmb{X}}^{(B)})\in \chi_{\alpha}^B|\pmb{Y},\pmb{Z})$. This  expression  can be decomposed into the sum of $P((\acute{\pmb{X}},\acute{\pmb{X}}^{(1)},\ldots ,\acute{\pmb{X}}^{(B)})\in \chi_{\alpha}^B|\pmb{Y},\pmb{Z})$ and $P((\pmb{X},\widetilde{\pmb{X}}^{(1)},\ldots ,\widetilde{\pmb{X}}^{(B)})\in \chi_{\alpha}^B|\pmb{Y},\pmb{Z})-P((\acute{\pmb{X}},\acute{\pmb{X}}^{(1)},\ldots ,\acute{\pmb{X}}^{(B)})\in \chi_{\alpha}^B|\pmb{Y},\pmb{Z})$. On one hand, the exchangeability of $B+1$ triples $(\acute{\pmb{X}}, \pmb{Y}, \pmb{Z}),(\acute{\pmb{X}}^{(1)}, \pmb{Y}, \pmb{Z}),\cdots,(\acute{\pmb{X}}^{(B)}, \pmb{Y}, \pmb{Z})$ under $H_0$, conditioned on $\pmb{Y}$ and $\pmb{Z}$, leads to the first term being smaller than $\alpha$. On the other hand, $(\acute{\pmb{X}}^{(1)},\ldots ,\acute{\pmb{X}}^{(B)})$, conditioned on $\acute{\pmb{X}},\pmb{Y}$ and $\pmb{Z}$, is generated from the same mechanism as $(\widetilde{\pmb{X}}^{(1)},\ldots ,\widetilde{\pmb{X}}^{(B)})$, conditioned on $\pmb{X},\pmb{Y}$ and $\pmb{Z}$. In other words, for all $\pmb{x}\in \mathbb{R}^n$, $((\widetilde{\pmb{X}}^{(1)},\ldots ,\widetilde{\pmb{X}}^{(B)})|\pmb{X}=\pmb{x},\pmb{Y},\pmb{Z})\overset{\mathrm{d}}{=}((\acute{\pmb{X}}^{(1)},\ldots ,\acute{\pmb{X}}^{(B)})|\acute{\pmb{X}}=\pmb{x},\pmb{Y},\pmb{Z})$. By Lemma 6 in the Supplementary Materials, we can bound the second term by the total variation distance between $\widehat{p}(\cdot|\pmb{Z})$ and $p(\cdot|\pmb{Z})$.
>
> Third, we discuss the non-trivial points in the power analysis of our method in Theorem 4. We utilize the Markov inequality to establish a connection between power analysis and the consistency of the $\mbox{CMI}$ estimator.  Mukherjee et al. (2020) considered the scenario where $p(x,z)p(y|z)$ (denoted as $g$) is known for the consistency.  However, this distribution is  unknown in practice.  We investigate the consistency  when incorporating the  $1$-NN sampling (Algorithm 1) to approximate it. In this extension, we denote the joint density of $(X,Y',Z)$ generated through $1$-NN sampling as $\phi$. The CMI estimator $\widehat{\mbox{CMI}}$ based on Algorithm 2 is referred to as $\widehat{D}\_{KL}^{(n)}(f||\phi)$, and the true CMI of $(X,Y,Z)$ is denoted as $\mbox{CMI}:=I(X;Y|Z)=D\_{KL}(f||g)$. The key aspect of this extension involves decomposing $\widehat{D}\_{KL}^{(n)}(f||\phi)-D\_{KL}(f||g)$ into two components: $\widehat{D}\_{KL}^{(n)}(f||\phi)-D\_{KL}(f||\phi)$ and $D\_{KL}(f||\phi)-D\_{KL}(f||g)$. The first term is shown to be $o\_p(1)$ according to Mukherjee et al. (2020). By leveraging properties of point-wise minimizers of binary cross-entropy loss, Jensen's inequality, and Lagrange's mean value theorem, we bound the second term by $d\_{TV}(\phi,g)$, which is also $o(1)$. The combination of these two results leads to the conclusion that $\widehat{D}\_{KL}^{(n)}(f||\phi)-D\_{KL}(f||g)=o_p(1)$, which confirms the consistency of our CMI estimator.
>
> **Question 1**:
> Please refer to the global response for details regarding the question "Why is the CMI estimated using classifiers rather than directly optimising (4) like f-GAN? Why not directly use the maximum of (4) to approximate the CMI?".

---

> > ### Comment · Reviewer_xTib · 2023-08-14
> > **Thanks for Replying**
> >
> > > Nevertheless, the MLMI-based method heavily depends on selecting appropriate basis functions that can effectively capture the information in Z, and it can face challenges due to the curse of dimensionality (Mukherjee et al., 2020).
> >
> > True. Thanks for clarifying.
> >
> > > Second, we discuss the non-trivial points in the type-I error control
> >
> > Thanks! These all make sense to me.
> >
> > I hope in the revision, the authors can explain how their results are linked with assumptions 1 and 2 (highlighting the necessity of smoothness in this context).
> >
> > In response to the satisfactory response to Weekness 1 and 3, I will raise my score by 1.

---

> > > ### Author Response · Authors · 2023-08-15
> > > **Thank  Reviewer xTib**
> > >
> > > Dear Reviewer xTib,
> > >
> > > Thanks for your positive feedback. We will incorporate your suggestions into the paper during the revision process.
> > >
> > > Thanks,
> > >
> > > Authors

---

### Official Review · Reviewer_BxQV · 2023-07-26

**Soundness:** 3 good
**Presentation:** 3 good
**Contribution:** 3 good
**Rating:** 5
**Confidence:** 3

**Summary:**

The paper proposes a novel method to test conditional independence (CI) by estimating conditional mutual information (CMI). A kNN local sampling strategy is employed to empirically sample from an unknown conditional distribution. The method is proved to have type-I error controlled, and power converging to 1.

The main contribution of the novel method consists of three points. First, it addresses challenges posed by high-dimensional conditioning variables and limited data samples. Second, it operates without assumptions about distribution forms or feature dependencies. Third, it avoids dataset splitting. Empirically, it is computationally efficient.

**Strengths:**

(1) Good originality. Using kNN as a sampling method to circumvent prior knowledge on conditional distribution $p_{X|Z}(x|z)$ is a novel idea.

(2) The algorithm is a high-quality one, in terms of enhanced computational efficiency.

(3) The simulation settings is a nice touch to showcase that the novel method operates without assumptions about distribution forms.

**Weaknesses:**

(1) The paper claims an improvement upon the current literature under high-dimensional settings. But this contribution is not reflected in either theoretical results or simulations. Specifically, in simulation the sample size is 500, while dimension of $Z$ is at most 100. The results are still in low-dimensional regime.

**Questions:**

(1) The paper claims novelty in avoiding sample splitting. It seems to me that Algorithm 1 & 2 are still splitting data into $V_1$ and $V_2$. Does this splitting not weaken test power?

(2) More runtime analysis would be appreciated. The paper's claim on enhanced computational complexity sounds legit, but requires more evidence to support.

(3) How does $\beta$ and $c_{d_Z}$ in Assumptions 1 and 2 affect the convergence rates in Theorems 2 and 4?

(4) Line 180: "Furthermore, it intuitively suggests that our test can achieve high power under $H_1$". This claim was based on $\hat{\text{CMI}}$ should be positive whp. This quantity, however, is replaced by $\tilde{\text{CMI}}^{(b)}$ in practice. It the latter still positive whp?

**Limitations:**

The paper claims that their novel method operates without assumptions about distribution forms or feature dependencies. However, the two assumptions on smoothness do not sound mild.

---

> ### Author Rebuttal · Authors · 2023-08-06
>
> We thank the reviewer for the helpful review!
>
> **Weakness**: Please refer to the global response for details regarding the claim “ high-dimensional conditioning variables".  Further, we conduct additional experiments to assess the performance of our tests in higher-dimensional scenarios. We generate simulation dataset based on `Scenario I: The post-nonlinear model'  in the paper. We keep the distribution as the standard Gaussian, maintain the sample size of $n=500$, and set the dimension of $Z$ to be 200 and 500. We present the type I error rate and testing power in Table 1 in the pdf file attached in the global response, where the results are averaged over 100 independent trials. It is shown that our test maintains good control of the type I error and achieves high testing power. Notably, LPCIT is  very time-consuming when the dimension of $Z$ is high, we thus do not report the results of this test.
>
> **Question 1**.  Please refer to the global response for details regarding the claim “our method avoids sample splitting".
>
> **Question 2**. First, we provide the total runtime and the time taken for a single test run for all methods on the real ABALONE dataset ($n=4177$) and the Flow-Cytometry dataset ($n=1755$).  Table 2 in the pdf file attached in the global response shows that our test maintains a high level of computational efficiency even when handling large sample sizes (e.g., $n=4177$). Second, please refer to Lines 152-158 in the paper for more details. Third, we find that our method has a computational complexity of $O(B\cdot dz \cdot n\log n)$. Further comparisons of computational complexity with other methods will be conducted in our future research.
>
> **Question 3**. In Theorem 2, $\beta$ is employed to establish an upper bound on the KL divergence between $p(x|Z)$ and $p(x|Z\_n^{(l)})$: $$D_{KL}\\{p(x|Z)||p(x|Z\_{n}^{(l)})\\}=\frac{1}{2}(Z_{n}^{(l)}-Z)^{T}I_{a}(Z)(Z_{n}^{(l)}-Z),$$
> where $0\leq \lambda_{\max}(I_{a}(z))\leq \beta$. Therefore, given that $\beta$ is finite, the convergence rate of $d\_{TV}\\{p(x|Z),\widehat{p}(x|Z)\\}$ is determined solely by the convergence rate of $\Vert Z\_{n}^{(l)}-Z\Vert\_{2}^{2}$ and remains unaffected by $\beta$, as indicated by the inequalities:
> \begin{align*}
>  d\_{TV}\\{p(x|Z),\widehat{p}(x|Z)\\}  \leq \sqrt{\frac{D_{KL}\\{p(x|Z),\widehat{p}(x|Z)\\}}{2}},\\
> D\_{KL}\\{p(x|Z),\widehat{p}(x|Z)\\} = \sum\_{l=1}^{k} I\\{\xi = l\\} D\_{KL}\\{p(x|Z)||p(x|Z\_{n}^{(l)})\\}.
> \end{align*}
>
> In the context of power analysis as presented in Theorem 4, it is noteworthy that most CIT methods lack theoretical results pertaining to testing power. Examples include works by Runge et al. (2018), Berrett et al. (2020), and Li et al. (2023). Even when theoretical analyses of testing power are conducted, they never give the convergence rate of testing power. For instance, the method proposed by Shi et al. (2021) achieves consistency only against a subset of alternatives in $H_1$. Wang et al. (2022) specifically consider the high-dimensional linear model and analyze the asymptotic power under local alternatives.
>
> In our work, we achieve the consistency of our test against all alternatives stated in $H_1$ when analyzing testing power. This consistency heavily relies on the consistency of the CMI estimator. The CMI estimator's consistency hinges on whether the joint density of $(X,Y',Z)$ generated by the 1-NN sampling (Algorithm 1), denoted as $\phi$, approximates $p(x,z)p(y|z)$ (denoted as $g$) well, in terms of TV distance. To bound the TV distance between $\phi$ and $g$, we utilize $\beta$ and $c\_{d_z}$. Precisely, $d_{TV}(\phi,g)\leq b(n)$, where
> $$b(n)=\frac{1}{2}\sqrt{\frac{\beta}{4} \frac{C_5 2^{1/d_z} \Gamma (1/d_z)}{(n\gamma_{d_z})^{1/d_z}d_z}+\frac{\beta \epsilon_1 G(2c_{d_z}\epsilon_1^2)}{4}}+\exp \bigg{(}-\frac{1}{2}n\gamma\_{d_z}c\_{d_z}\epsilon_1^{d_z+2}\bigg{)}+G(2c\_{d_z}\epsilon_1^2),$$
> where $\epsilon_1$ is small enough, $\Gamma(\cdot)$ is the gamma function, and $\gamma_{d}$ is the volume of the unit radius $l_2$ ball in $\mathbb{R}^d$. As indicated in Theroem 1 of Sen et al. (2017), as long as $\beta$ and $c\_{d_z}$ are finite (Gao et al., 2017),  $d\_{TV}(\phi,g)$ can be small enough when $n$ is large enough. This ensures the consistency of our test against all alternatives stated in $H_1$.
>
> [1] Wang W., Janson, L. A high-dimensional power analysis of the conditional randomization test and knockoffs, Biometrika, 2022.
>
> **Question 4**.  We have proved that $\widehat{\mbox{CMI}}$ is a consistent estimator of $\mbox{CMI}$  based on the original data set $(\pmb{X},\pmb{Y},\pmb{Z})$. Because the  $\mbox{CMI}$ is greater than zero under $H_1$, $\widehat{\mbox{CMI}}$ is positive whp.
>
> When the distribution of $X|Z$ is known, we can  draw  pseudo samples $\pmb{X}^{(b)}$. In this case,  the estimator $\widetilde{\mbox{CMI}}^{(b)}$ based on $(\pmb{X}^{(b)},\pmb{Y},\pmb{Z})$ converges to zero, but it is not necessarily positive. As a result, the $p$-value calculated using $p:=\\{1+\sum_{b=1}^{B}{1}(\widetilde{\mbox{CMI}}{}^{(b)}\geq \widehat{\mbox{CMI}})\\}/(1+B)$ is highly likely to be very small, which indicates the consistency of our test against all alternatives stated in $H_1$.
>
> In practice, the distribution of $X|Z$ is  unknown.  We use the $k$-NN local sampling strategy (Algorithm 3) to draw pseudo samples $\widetilde{\pmb{X}}^{(b)}$. In the proof of Theorem 4, we have demonstrated that $\widehat{\mbox{CMI}}^{(b)}$ calculated based on $(\widetilde{\pmb{X}}^{(b)},\pmb{Y},\pmb{Z})$ converges to zero, though it is not necessarily a positive value. Consequently, the $p$-value calculated using $p:=\\{1+\sum_{b=1}^{B}{1}(\widehat{\mbox{CMI}}{}^{(b)}\geq \widehat{\mbox{CMI}})\\}/(1+B)$ is very small with high probability, indicating our test can achieve high power under $H_1$.
>
> **Limitation**: Please see the global response for details regarding  the claim “our method operates without assumptions about distribution forms or feature dependencies".

---

> > ### Comment · Reviewer_BxQV · 2023-08-19
> >
> > I appreciate the clarification from the authors. All my questions are answered.
> >
> > The weakness that I raised seems to be well explained. I'll raise my score by 1.

---

> > > ### Author Response · Authors · 2023-08-20
> > > **Thank Reviewer BxQV**
> > >
> > > Dear Reviewer BxQV,
> > >
> > > Thanks for your positive feedback.
> > >
> > > Authors

---

### Author Rebuttal · Authors · 2023-08-06

We thank the reviewers for their detailed and considerate feedback. We offer the following clarifications.  **A PDF file including 5 tables is attached.**

1. About the claim "**our method avoids sample splitting**", we  emphasize that
our proposed method  **allows the entire dataset to be used in computing
$\widehat{\mbox{CMI}}$**. The method in Li et al. (2023) requires splitting the dataset into two parts, and thus the dataset used for calculating the test statistics comprises only one-third of the total samples. This will reduce the statistical power of the  test, particularly when working with small datasets. In contrast, our proposed procedure  **allows the entire dataset to be used in computing $\widehat{\mbox{CMI}}$**, thereby avoiding the loss of testing power. In the computation of the CMI estimator using the entire dataset, we adopt a classifier-based approach. This approach necessitates the utilization of two datasets for a  supervised classification task. As a result, the data splitting in this context occurs naturally and does not result in any loss of testing power.

2. Regarding the claim  "**high-dimensional conditioning variables**", we have the following response. Recently, Cai et al.(2022), Kim et al. (2022), and Ai et al. (2022) have primarily concentrated on low-dimensional cases where the dimension of $Z$ does not exceed 2. Additionally, Runge (2018) employed a sample size of 500, while restricting the dimension of $Z$ to a maximum of 8. CI testing is a really challenging task when handling high-dimensional conditioning variable sets (Scetbon et al., 2022). The concept "high dimensional" in CI testing refers to the scenario where the dimension of $Z$ is not low or is relatively high given the sample size (Scetbon et al., 2022). This concept is different from that of the high-dimensional regression, where some sparsity assumptions may be needed.

For example, Sen et al. (2017) and Scetbon et al. (2022) both claimed they can handle the high-dimensionality of $Z$. In their simulation studies, Sen et al. (2017) set the sample size to be 1000 with the dimension of $Z$ being at most 150, and Scetbon et al. (2022) set the sample size to be 1000 with the dimension of $Z$ being at most 50. They also did not consider that the dimension of $Z$ converges to infinite with the sample size in their theoretical results. In Figure 1 of our paper, we present the favorable performance of our tests for $n=300$ and the dimension of $Z$ equal to 80. In response to Reviewer BxQV, we also consider $Z$ dimension as high as $500$ when $n=500$. In contrast to previous works, the dimension of $Z$ is relatively high.

[1] Cai et al., A Distribution Free Conditional Independence Test with Applications to Causal Discovery, JMLR 2022.

[2] Ai et al., Testing Unconditional and Conditional Independence Via Mutual Information, Journal of econometrics 2022.

3.  Regarding the claim "**our method operates without assumptions about distribution forms or feature dependencies**", we emphasize that, unlike Candes et al. (2018), our method operates successfully in practical applications without requiring the true distribution of $X|Z$ to be provided. We propose using the k-nearest-neighbor local sampling strategy to approximate  $p\_{X|Z}(x|z)$.  We establish in Section 4 that the total variation distance between the true distribution of $X|Z$ and the distribution of samples generated by the k-nearest-neighbor local sampling strategy tends to zero in probability as $n$ goes to infinity.

However, in the theoretical part, we need Assumptions 1 and 2 to facilitate our proof. The two assumptions  have been included in Gao et al. (2016, 2017) and Sen et al. (2017), even though they may not be the weakest conditions. Moreover, we can validate Assumption 1 when ($X, Z$) follows the multivariate Gaussian distribution  (MVD) and Assumption 2 when $Z$ follows the MVD.

4. About the question "**Why is the CMI estimated using classifiers rather than directly optimising (4) like f-GAN? Why not directly use the maximum of (4) to approximate the CMI?**", we have the following response. Belghazi et al. (2018) introduced a mutual information neural estimator (MINE) by maximizing equation (4) directly. We obtained the conditional version of this estimator to estimate CMI, referred to as C-MINE. This estimator does not require dataset splitting into training and testing sets. However, through simulations, we observed that C-MINE has a high variance and its optimization becomes unstable in high-dimensional settings. Similar conclusions have been drawn by Poole et al. (2019) and Mukherjee et al. (2020). Furthermore, C-MINE exhibits more computational complexity compared to our classifier-based CMI estimator. As outlined in Algorithm 3, to obtain a single p-value, we need to estimate $B+1$ CMI values. Therefore, any method that can offer fast and accurate estimation will yield substantial benefits.

Furthermore, we note that  Mondal et al. (2020) directly optimized (4) like f-GAN to estimate CMI. However, the training of GANs is often  challenging, with the risk of collapse if hyperparameters and regularizers are not carefully chosen (Dhariwal and Nichol, 2021). In contrast, our classifier-based CMI estimation method relies on a simple binary classifier and performs effectively even in high-dimensional scenarios. Importantly, our classifier-based approach demonstrates remarkable computational efficiency compared to the direct optimization of equation(4) by f-GAN.

[1] Belghazi et al., Mutual Information Neural Estimation, ICML 2018.

[2] Poole et al., On Variational Bounds of Mutual Information, ICML 2019.

[3] Kumar Mondal et al., C-MI-GAN: Estimation of Conditional Mutual Information Using MinMax Formulation, UAI 2020.

[4] Prafulla Dhariwal and Alex Nichol, Diffusion Models Beat GANs on Image Synthesis, NeurIPS 2021.

---

### Decision · Program_Chairs · 2023-09-21

**Decision:**

Accept (poster)

**Comment:**

The paper presents a novel approach to Conditional Independence testing, leveraging kNN as a sampling method, which enhances computational efficiency. The work is theoretically sound, backed up well by theorems, and the algorithm is straightforward to implement. The paper is well-written and sufficiently readable.  While there are some concerns about the extent of its advance to the existing works, these do not overshadow the paper's strengths. Therefore, the recommendation is to accept the paper.